# BuckTales: A multi-UAV dataset for multi-object tracking and re-identification of wild antelopes

**Hemal Naik**[1,2,3,4*]       **Junran Yang**[1]       **Dipin Das**[1]

**Margaret C Crofoot**[1,2,3]       **Akanksha Rathore**[1,2,4,5]

**Vivek Hari Sridhar**[1,2,3,4]

Department for the Ecology of Animal Societies, Max Planck Institute of Animal Behavior[1]
Center for the Advanced Study of Collective Behaviour, University of Konstanz[2]
Department of Biology, University of Konstanz[3]
Centre for Ecological Sciences, Indian Institute of Science[4]
Computer Science & Information Systems, BITS Pilani, Hyderabad[5]

## Abstract

Understanding animal behaviour is central to predicting, understanding, and mitigating impacts of natural and anthropogenic changes on animal populations and ecosystems. However, the challenges of acquiring and processing long-term, ecologically relevant data in wild settings have constrained the scope of behavioural research. The increasing availability of Unmanned Aerial Vehicles (UAVs), coupled with advances in machine learning, has opened new opportunities for wildlife monitoring using aerial tracking. However, limited availability of datasets with wild animals in natural habitats has hindered progress in automated computer vision solutions for long-term animal tracking. Here we introduce *BuckTales*, the first large-scale UAV dataset designed to solve multi-object tracking (MOT) and re-identification (Re-ID) problem in wild animals, specifically the mating behaviour (or lekking) of blackbuck antelopes. Collected in collaboration with biologists, the MOT dataset includes over 1.2 million annotations including 680 tracks across 12 high-resolution (5.4K) videos, each averaging 66 seconds and featuring 30 to 130 individuals. The Re-ID dataset includes 730 individuals captured with two UAVs simultaneously. The dataset is designed to drive scalable, long-term animal behaviour tracking using multiple camera sensors. By providing baseline performance with two detectors, and benchmarking several state-of-the-art tracking methods, our dataset reflects the real-world challenges of tracking wild animals in socially and ecologically relevant contexts. In making these data widely available, we hope to catalyze progress in MOT and Re-ID for wild animals, fostering insights into animal behaviour, conservation efforts, and ecosystem dynamics through automated, long-term monitoring.

## 1 Introduction

In a rapidly changing world, behaviour is the most flexible tool in an animal's toolkit to solve problems and adapt to novel challenges. While numerous observational and automated sensor-based techniques exist to study animal behaviour, the past decade has seen a significant increase in the

---

*Corresponding authors: hmnaiks@gmail.com, vivekhsridhar@gmail.com

38th Conference on Neural Information Processing Systems (NeurIPS 2024) Track on Datasets and Benchmarks.

use of animal tracking from camera-based technology [Couzin and Heins, 2023, Tuia et al., 2022]. The ability to record detailed movement and behavioural data of individual animals non-invasively, and over extended periods, has facilitated a range of discoveries across various aspects of animal lives—from the study of animal architecture [Smith et al., 2021] and the understanding of individual and collective animal decision-making [Sridhar et al., 2021, Sampaio et al., 2024], to revealing the mechanisms of social interactions that result in coordinated collective movement [Torney et al., 2018, Sridhar et al., 2023] and anti-predatory strategies [Rosenthal et al., 2015, Sosna et al., 2019, Davidson et al., 2021].

While insights gained from vision-based animal tracking are on the rise, a lot of this work is still restricted to controlled laboratory settings [Naik et al., 2023]. To translate these methods for research on animals in their natural habitats, biologists are beginning to leverage advances in aerial imagery (through the use of Unmanned Aerial Vehicles or UAVs) and state-of-the-art techniques in computer vision and machine learning to record and track freely-moving animals in the wild [Koger et al., 2023, Price et al., 2023, Ozogány et al., 2023, Maeda et al., 2021]. However, the problem remains that most publicly available datasets of animal monitoring from UAVs do not offer ground truth for movement tracking problems. Furthermore, animals in the wild do not restrict their movement to specific areas captured within field-of-view of a single UAV. Therefore, recording behaviours occurring in relatively large spatial scales requires robust tracking solutions and the fusion of information from multiple simultaneously-flying UAVs.

Here we present BuckTales, the first dataset designed to study the behaviour of wild animals (in this case, blackbuck) in a large area using multiple simultaneously-flying UAVs. The dataset focuses on the task of UAV-based multi-object tracking (MOT) and re-identification (Re-ID) of wild animals. The MOT dataset contains 22.5K frames (12.5 min) in total with the longest video >3 min (5805 frames) and average of 75 individuals per video. The Re-ID dataset focuses on the task of merging the tracking data from multiple UAVs and thus annotations are provided for each antelope moving in the overlapping region of a UAV pair. These data are collected in collaboration with behavioural biologists studying lekking–an extremely rare mating system observed in <2% of mammalian species. Thus, our dataset presents a unique opportunity where results from algorithms trained here may have cross-disciplinary and direct real-world applications.

This article is written for both biologists and computer scientists. Therefore, we provide details for creating such datasets and the relevant code base for biologists to replicate our process. Similarly, for the computer science community, we offer baseline experiments with commonly known detection and tracking methods to demonstrate limitations of state-of-the-art methods. An added contribution of our dataset is that it is accompanied by specific analyses that highlight its suitability to becoming a benchmark dataset for solving large scale animal tracking with multiple camera sensors.

## 2    State of the Art

It is hard to capture videos of detailed animal behaviour in the wild. Readying a dataset that captures diverse animal behaviours requires extensive planning, logistical and financial support, and sizable human effort to record and annotate massive amounts of images and videos. With that in mind, computer scientists often turn to annotating images and videos widely available on the internet [Chen et al., 2023]). While models trained on these datasets have greatly furthered our ability to detect animals under diverse backgrounds and environments, the methods themselves have limited portability to solve problems encountered by biologists. On the other hand, biologists capture large scale datasets of wild animals in ecologically relevant settings but invest little to create annotated datasets to facilitate method development by computer scientists. The lack of collaboration between the two fields has led to a dearth of high-quality datasets, resulting in few researchers working on challenging computer vision problems relevant to wildlife. In this section, we showcase the scarcity of datasets by highlighting the state-of-the-art datasets involving wild animals captured with UAVs in context of multi-object tracking (MOT) and re-identification (Re-ID). Additionally, we also point out relevant contributions of this paper.

### 2.1    Multi-object tracking with UAVs

Creating a good multi-object tracking dataset with wild animals is challenging. There are no standards for knowing what comprises a good MOT dataset. Various aspects like i) having long annotated

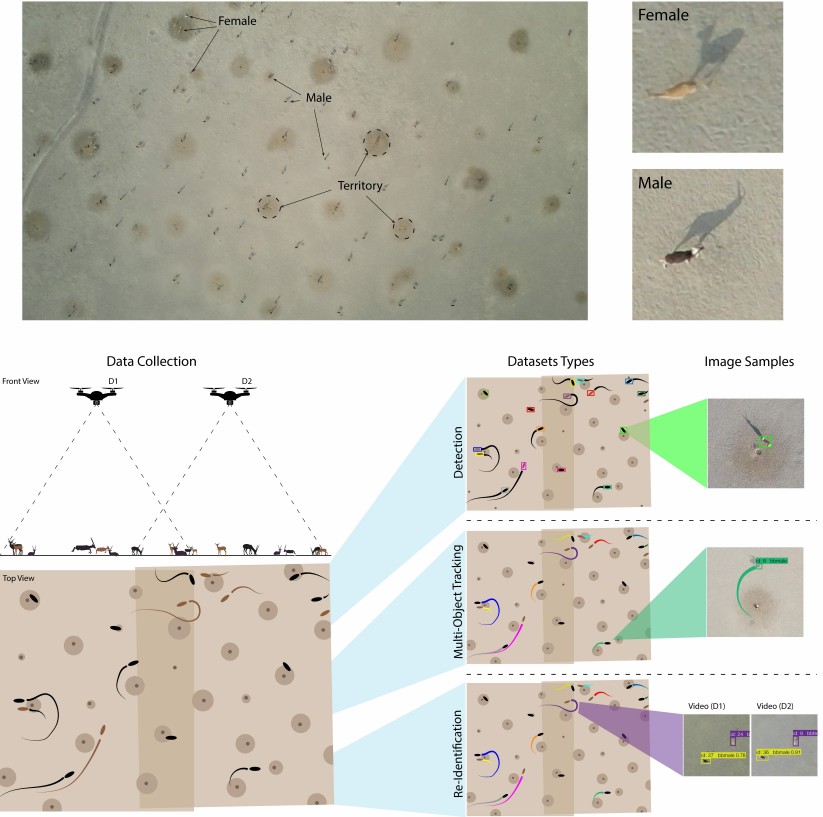

Figure 1: A schematic of the data collection strategy and dataset details. The image in the top displays top-down view of a blackbuck lek from a single drone with male territories marked. The close-ups on the right show an example male and female. The bottom figure is a simplified data collection scheme, shown here with two drones (note that the actual collection scheme involved three drones). Three types of annotations are made available with the manuscript: object detection, multi-object tracking and re-identification.

sequences with multiple moving targets, ii) capturing variation in animal movement and behavioural characteristics, and iii) encompassing varied proximity to neighbours during different types of social interactions, all appear to be important. With that in mind, scraping videos off the internet is far from optimal as it does not guarantee that the dataset will account for the above-mentioned diversities essential for a good MOT dataset.

A predominant approach to the MOT problem is tracking by detection. Perhaps, this is why most UAV datasets of wild animals largely focus on the object detection problem (see Table 2). Even within the biological literature, researchers have proposed tracking by detection to track wild animals with UAVs [Koger et al., 2023, Rathore et al., 2023b]. The problem however persists that these (and other) approaches can not be evaluated for their tracking accuracy due to the lack of a benchmark dataset for multi-object tracking of animals in the wild. [Price et al., 2023] recently offered a fast method for annotating datasets for the task of automated behaviour recognition. The method makes use of multi-object tracking to find individuals in videos but the authors do not provide datasets with annotations for the MOT problem. [Bonetto and Ahmad, 2023] presented a novel idea using synthetic data to improve animal detection, thus reducing the need for large real-world datasets. However, such synthetic models for animal movement cannot reflect real movements of animals and thus have limited contribution towards the MOT problem.

One of the only animal MOT datasets with UAVs consists of cows grazing on a farmland [Barrios et al., 2024]. The dataset offers a total of 106 seconds of annotated data split into 7 video sequences. Here, the main intention was to develop a method for tracking cattle with thermal cameras and the dataset contains relatively short segments which are not ideal for evaluating tracking performance

over longer periods of time. Other MOT datasets include, terrestrial cameras in outdoor setting such as AnimalTrack [Zhang et al., 2023], semi-controlled [Waldmann et al., 2024] or controlled settings [Naik et al., 2023]. Generic MOT datasets such as TAO dataset ([Bai et al., 2021]) and GMOT-40 [Dave et al., 2020] do contain some sequences with animals but arguably, the challenges of solving MOT from an aerial view are different and require specialised datasets that address this niche.

Finally, it should be noted that MOT datasets with UAVs designed for tracking humans or man-made objects in urban areas are also scarce [Zhu et al., 2021]. We provide some of the longest annotated sequences (>3 min) at the highest resolution (5.4K) and show (with further analysis) that longer trajectories allow greater opportunities to evaluate the robustness of tracking as multiple individuals interact with each other. Our dataset contains tracking challenges which directly contribute towards improving existing MOT solutions irrespective of the target object.

## 2.2 Re-identification with UAVs

The problem of identity tracking is highly relevant in the context of long-term monitoring of animals. Re-identification is largely pursued with the intention of monitoring the same individuals between different days or even years. Image-based individual identification between multiple days is successfully demonstrated with some animals [Wahltinez and Wahltinez, 2024] in the wild but it is largely practiced in controlled environment such as lab studies [Romero-Ferrero et al., 2019, Walter and Couzin, 2021]. Re-identification algorithms use unique body patterns (whales [Berger-Wolf et al., 2017],tigers [Li et al., 2019], cow [Li et al., 2022]) or face recognition like algorithms to assign identities e.g. chimps [Schofield et al., 2019, Paulet et al., 2024].

More recently, UAVs are also being deployed for individual identification in animal behaviour research [Ozogány et al., 2023, Maeda et al., 2021]. To the best of our knowledge, only two datasets are available for Re-ID of animals using UAV images. The first is a datasets with 88,000 images of 139 crocodiles [Desai et al., 2022] and the second is 706 images of 10 cows [Mücher et al., 2022]. Animals like blackbucks do not exhibit visibly unique patterns and thus creating a ground truth with human annotators is extremely challenging, even with terrestrial cameras (e.g. camera traps). In this paper, we define the problem of re-identification as a problem of identifying the same individual recorded at the same time (Fig 2) with two separate UAVs. This approach does not require human annotators to identify individuals specifically and ground truth can be created by establishing identities between simultaneously recorded videos (see Sec. 3.3.3). A similar approach was used for Re-ID of cars by [Wang et al., 2019] for the purpose aerial surveillance using multiple UAVs but such datasets do not exist for monitoring wild animals.

Our dataset is the first to tackle the Re-ID problem for a large group of animals using multiple UAVs and the first to provide video annotations for the Re-ID problem. We have defined the Re-ID problem in context of combining tracking data obtained from multiple sensors and this approach paves the way for solving large scale monitoring with a fleet of automated UAVs in the wild.

# 3 BuckTales dataset

This section contains details about the conceptualization of the dataset, the data collection protocol and the structure of the dataset. First, we inform the reader about lekking behaviour of blackbuck. Then we offer information about our data collection scheme which establishes relevance to the multi-object tracking and the Re-ID problem. The final section provides structure of the dataset along with the annotation scheme used for preparing the dataset.

## 3.1 Behaviour

Lekking is a relatively rare mating system in which males aggregate in closely clustered territories that females visit to find mates [Rathore et al., 2023a]. Both the aggregation, and the site of its occurrence, are known as leks. Fig 1 shows an aerial image and a schematic of a lek. The circular spots seen in the lek are territorial markings of males. Males often engage in agonistic interactions, including ritualistic displays, fights and chases, with their neighbours. When females visit a territory, their interactions with the territory-owner can vary from courtship displays to mounting in a few seconds to several hours. Females also move across territories assessing different males while being chased, displayed to, and courted. With a wide array of social interactions in an ecologically and

evolutionarily relevant context, leks serve as an excellent case study to address with multi-object tracking (MOT).

## 3.2 Data collection scheme

Videos of the lek were obtained at resolution of 5.4K and a frame rate of 30 fps with all UAVs flying at an altitude of 80 metres. This recording altitude was chosen based on previous UAV-based studies on blackbuck [Rathore et al., 2023b] as it provides a balance between a broad field of view and the number of pixels per animal. Given the size of the lek (∼250 m x 300 m), we employed three DJI Air 2S drones simultaneously to cover a majority of the lekking arena. All recordings were obtained with the cameras pointing at -90°(directly downward) to minimise occlusions. Fig 1 shows a schematic representation of our data collection strategy.

In total, we collected ∼10 TB of data, consisting of video sequences lasting between 30 and 120 minutes. These extended-duration recordings were achieved using a relay technique [Koger et al., 2023], detailed in the supplementary material. The dataset does not include images of blackbuck outside the lekking area, nor does it capture blackbuck herds, as the species typically moves in groups within the same habitat. Additionally, seasonal variations in coat color—such as the lighter coloration males exhibit outside the breeding season—are not reflected in the dataset. Due to these factors, we do not anticipate the misuse of our dataset in contexts like hunting or poaching.

## 3.3 Dataset structure and annotation process

We divided the dataset in three parts to focus on three different problems separately (See Fig 1). The first part consists of annotated images for object detection, the second part contains annotated videos for multi-object tracking, and the third part includes 11 pairs of videos from two simultaneously flying drones (with an overlap) for addressing the Re-ID problem.

### 3.3.1 Object detection dataset

Object detection is a necessary prerequisite for designing successful tracking algorithms, especially for the tracking by detection approach. However, videos with annotations for the MOT do not reflect sufficient variation in images across different days of lek and thus are not representative of the entire behavioural dataset. Therefore we prepared a separate dataset to build a reliable blackbuck detection algorithm. The intention is to use the same algorithm on the entire dataset collected for behavioural studies and the detection models trained on this dataset are used for evaluating different trackers on the MOT dataset (see Sec. 3.3.2). The detection dataset consists of 320 images selected from nine different days and a total of 18.4K annotations of male and female blackbuck (details supplementary). We use Labelbox to annotate the images with bounding boxes. The images are selected with careful consideration of factors such as time of the day (sunrise/sunset), the presence of shadows of the animals, the lighting conditions (weather) and most importantly, the presence of males and females on the lek.

### 3.3.2 Multi-object tracking (MOT) dataset

The MOT dataset focuses on the tracking of two class categories: blackbuck males and blackbuck females. All individuals are marked with a class label and unique individual IDs using the DarkLabel tool [2]. Each individual is manually selected in the first frame and DarkLabel automatically propagates the annotations to the next frame. The tool assigns a new bounding box to the individual while maintaining the identity and class information. A team of domain experts then verified the propagated annotations. We also wrote additional scripts to identify manual errors in the annotations e.g. ID duplications or missing class assignments (see supplementary). These scripts are provided with the dataset and will benefit others biologists who wish to replicate our approach.

We selected videos based on the variation in lekking activity. The MOT dataset includes 12 video sequences annotated with a total number of 1.2 million bounding boxes and they consist an average 60 tracks per video. Our longest video is 193 seconds (>5800 frames) and consists of 127 individuals. We highlight that continuous annotations of such magnitude are not available with any existing MOT datasets including captive or wild animals. Sec 4 contains more information on the quality of the

---

[2]https://github.com/darkpgmr/DarkLabel

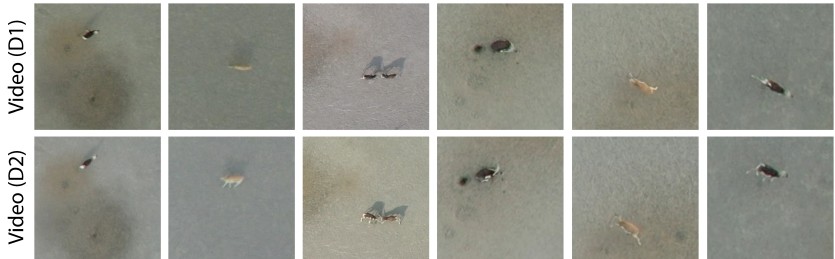

Figure 2: Sample images from the Re-ID dataset. The images in each column demonstrates the variation in appearance of an individual when captured from two different drones.

dataset w.r.t. the MOT problem and supplementary material includes the annotation details of each video.

### 3.3.3 Re-identification (Re-ID) dataset

Our data collection scheme involves multiple simultaneous drones with overlapping fields of view (see Fig 1). Since individuals are tracked independently in each video, a re-identification strategy is required to fuse data in the overlap between two adjacent video streams. This problem of data fusion is highly relevant for multi-sensor based studies, especially when data fusion is not possible with camera calibration or image stitching due to lack of significant overlap or unique matching features. Preparing ground truth for re-identification problem is extremely challenging in the case of animals that appear identical. Even field experts are unable to identify individuals blackbucks between different days. We used a machine learning enabled annotation workflow to simplify re-identification of same individuals from two different video streams with slightly different appearance of the animal. For annotation, we trained a YOLOv8 model using the detection dataset and used ByteTracker to assign tracking IDs to all individuals in each pair of videos. Annotators then watched the two videos augmented with IDs (provided with dataset) and marked IDs of all individuals visiting the overlapping areas by using movement to establish their identity. A summary file was then created with annotation of IDs from both videos mapped with corresponding frames numbers and their bounding box positions in raw videos. Figure 2 shows example images of difference scenarios demonstrating challenges within this dataset.

Our dataset is the first of its kind, offering ground truth for re-identification of individuals in a scenario where domain experts are unable to do so. The datasets consists of 730 tracks from 11 different video-pairs. The average track length of each video is between 600-1100 frames, and the duration of the tracks account for the movement of the animal in the overlapping region only (See suppl. video). It is possible to use the dataset for both one shot learning or temporal learning of individual features, especially in case of animals with similar appearance. The problem given in this paper is technically an extension of feature based re-identification method used for tracking occluded targets within the same video.

### 3.3.4 Data format & availability

The detection data is provided in the COCO and YOLOv8 format. The MOT annotations are provided in the MOT challenge format [3]. The MOT data is labeled using DarkLabel tool with a customized format and this is also provided with the dataset. The Re-ID dataset is provided in a customised format (details in supplementary). The dataset is publicly available as BuckTales at Edmond platform. Further, links for relevant repositories are provided in the supplementary.

## 4 Dataset analysis & benchmarking

Blackbuck, being moderately sized and among the fastest terrestrial mammals (capable of running up to 80 km/h), present a broad range of tracking challenges relevant to the study of animal behaviour.

---

[3]https://motchallenge.net/

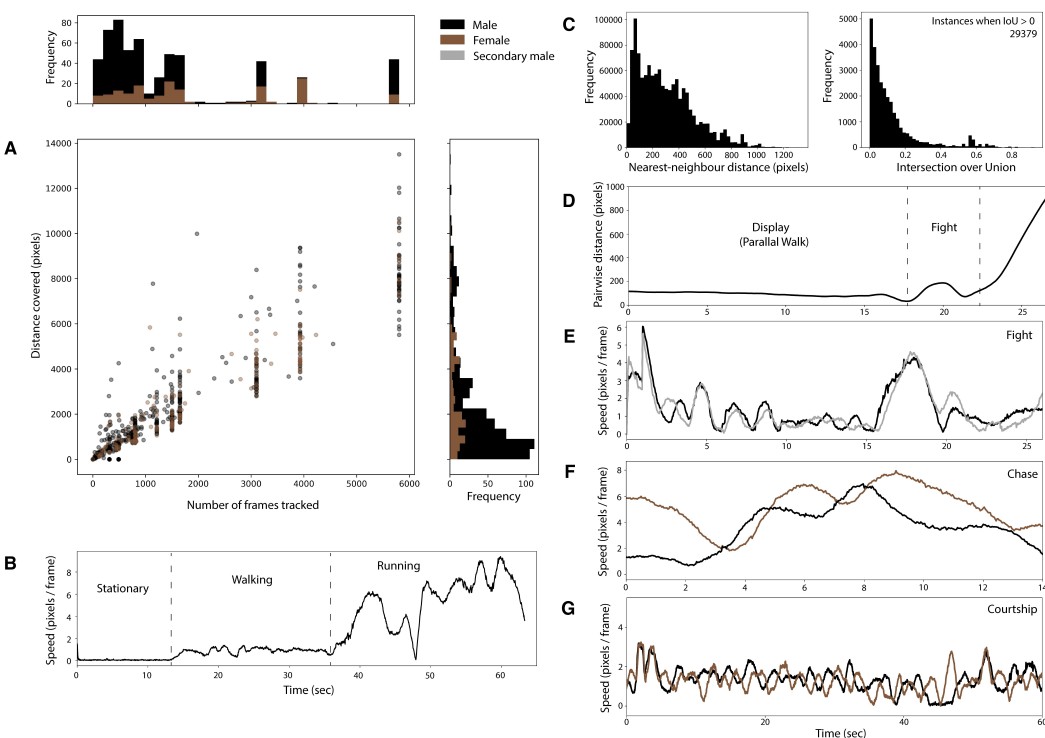

Figure 3: A cursory movement analysis highlighting the movement and behavioural diversity captured by our dataset. **A,B)** show diversity in individual specific behaviours while **C–G)** shows the same in pairs of interacting individuals.

Hence, effectively tracking these antelopes could generalize to other similarly sized or larger mammals. In this section, we provide a cursory analysis of the dataset and discuss characteristics relevant to the MOT problem. We also provide baseline results of two state-of-the-art object detection methods and benchmarking for MOT with three different tracking approaches.

## 4.1 Movement patterns in dataset

We combined all annotations from different videos to showcase common trends and patterns related to the movement of male and female blackbuck in the annotated videos. These analyses serve as indicators of the value this dataset adds with respect to MOT problem. Each video contains individuals exhibiting different types of behaviours i.e. resting, walking, running, displaying, courting, fighting etc. and signatures of these behaviours are visible in their movement characteristics. Below, we quantify distributions of various behaviours in which individuals engage and use pairwise distance and speed profiles to illustrate signatures of these behaviours in their movement.

**Length of annotations**: First, we computed the number of annotated frames per individual to see length profile of annotations. More than 90% individuals in our dataset are annotated for over 300 frames (>10 seconds), and over 50% individuals are annotated for more than 900 frames (>30 seconds). This variation in the tracking duration primarily occurs due to one of two reasons: i) Variation in the length of annotated video segments. The shortest video sequence contains 752 (∼25 seconds) frames while the longest video sequence contains over 5800 frames (>3 min). ii) Duration for which the animal stays within the frame. Even within video sequences, some animals do not remain in the video for the entire length of the video and thus are annotated for a shorter duration (see Fig 3A).

**Distance coverage**: In addition to the number of frames tracked, we measured each individual's distance covered in our dataset by measuring accumulated movement in pixels of bounding box center. (Fig 3A; see supplementary information for the same plot split by video). While there are more

males on the lek than females (68.7% males and 31.3% females), no obvious differences are noted in the distances they travel (Fig 3A). Our analyses show a wide range in the distance distribution of individuals and a strong correlation between the number of frames an individual is tracked and the distance it travels (Fig 3A). We show that individuals that travel less can in part be accounted for by the short length of their tracks, highlighting the need for longer-term tracking to gain biological insights from these types of data.

**Speed**: Another individual-specific metric we compute is speed. In order to highlight the various behaviours that can be captured by this kinematic measure, we present the speed profile (Fig 3B) of a sample individual that exhibits stationary, walking and running behaviours, all within a minute. These sudden transitions in individual speed profiles make movement-based tracking (for example using a Kalman filter) a challenge.

**Proximity**: After measuring individual movement metrics (distance covered and speed), we further quantified occasions where two individuals engaged in various types of interactions. As a first approximation of interactions, we quantified proximity between individuals by calculating two different metrics: i) The distribution of distances of each individual to its nearest neighbour. ii) IoU (Intersection over Union) of individual bounding boxes (Fig 3C). Combining insights from these two measures, we show that most individuals have at least one other individual that is within 200 pixels and that our dataset includes ∼29k instances where at least two individuals have overlapping bounding boxes. These moments of close proximity capture various inter-individual interactions like displays, courtship and fighting (see supplementary). While these behaviours represent parts of the data that are of most interest to biologists, they also comprise moments with an increased probability of ID switching and loss of individual tracking.

**Inter-individual interactions**: Finally, we also visualise movement characteristics i.e., distance between or speed profiles of, pairs of individuals when they engage in the above-mentioned close proximity behaviours (Fig 3D–G). Fig 3D shows the distance between two males displaying to each other (also referred to as a parallel walk). Figs 3E–G show speed profiles of i) two males fighting (Fig 3E), ii) a male chasing a female (Fig 3F), and iii) a male courting a female (Fig 3G). Once again, these behaviours represent biologically interesting moments within these data but also represent moments when the tracking is most likely to fail.

## 4.2 Dataset Benchmarking

This subsection reports performance of the state-of-the-art detection and tracking algorithms on our dataset. We trained object detection models using the object detection dataset and used the best-performing detector with various tracking algorithms for benchmarking. Using a separate detection and tracking dataset allowed us to train a strong detector and leverage the entire MOT dataset for evaluation of the tracking results. We have not provided a baseline for the Re-ID problem.

### 4.2.1 Object detection

We trained detectors based on two state-of-the-art object detection models: Fast-RCNN (through Detectron2 [Wu et al., 2019]) and YOLOv8 [Jocher et al., 2023]. According to availability, NVIDIA A100 (40GB) and NVIDIA V100 (32GB) are used to train all our detection models. We trained models with different image resolutions and different class configurations. All details of the experiments are provided in the supplementary material with information of the train, validation, and test sets. We used mAP as metric for selecting the best detector.

Our results show that YOLOv8m with image size of 5472 obtains the best mAP score of 0.624. The detection dataset contains annotations for categories other than blackbuck i.e. birds, drones, unknown (dog) etc. Our experiments suggest that the model trained only with male and female class categories performs better than one with all classes included, probably due to the rare occurrence of the other categories in our dataset. Another notable observation is that YOLOv8m with a halved image resolution trains 10x faster with minimal impact on the mAP results (see supplementary), which can be a practical consideration for processing large scale dataset with negligible performance loss.

Table 1: Multi-object Tracking Benchmarking

| Method | Re-ID | HOTA$^\uparrow$ | DetA$^\uparrow$ | AssA$^\uparrow$ | MOTA$^\uparrow$ | MOTP$^\uparrow$ | IDF1$^\uparrow$ | IDsw$^\downarrow$ |
|---|---|---|---|---|---|---|---|---|
| ByteTrack | - | 0.4950 | 0.5124 | 0.4868 | 0.9347 | 0.5932 | 0.8335 | 777 |
| OC-SORT | - | 0.4203 | 0.4880 | 0.3726 | 0.8699 | 0.5928 | 0.6421 | 8736 |
| BoT-SORT | - | 0.5286 | 0.5165 | 0.5495 | 0.9429 | 0.5933 | 0.9214 | 171 |
|  | osnet_x0_25 | 0.5365 | 0.5159 | 0.5665 | 0.9430 | 0.5930 | 0.9418 | 158 |
|  | osnet_x1_0 | 0.5378 | 0.5160 | 0.5690 | 0.9429 | 0.5931 | 0.9451 | 153 |
|  | osnet_ain_x1_0 | 0.5369 | 0.5160 | 0.5671 | 0.9430 | 0.5931 | 0.9435 | 150 |
|  | lmbn_n | 0.5355 | 0.5159 | 0.5644 | 0.9427 | 0.5929 | 0.9400 | 169 |

#### 4.2.2 Multi-object tracking

We evaluated state-of-the-art MOT tracking methods such as ByteTrack [Zhang et al., 2021], OC-SORT [Cao et al., 2023], and BoT-SORT [Aharon et al., 2022]. Four different pre-trained Re-ID models, trained with human data, are used to make appearance based tracking with BoT-SORT (details A.4). We used the default configuration in the BoxMOT library [4] and evaluated our results using TrackEval[5]. All of these approaches belong to the tracking by detection paradigm and use the detections from our the best performing YOLOv8 model (see supplementary).

We present HOTA, MOTA, MOTP, and IDF1 metrics in Table.1, which are widely employed in the field as the main metrics to evaluate tracker performance. Our results show BoT-SORT as the best-performing tracker with the highest HOTA metric and the lowest number of ID switches. This shows that models with feature based (Re-ID) tracking are better than localization based tracking approaches. It should however be noted that Re-ID models require a dedicated GPU (in our case, the NVIDIA Quadro RTX 6000) whereas the other approaches do not require them. Our analysis highlights a known limitation of MOTA [Luiten et al., 2021], which overlooks association errors, potentially leading to artificially high scores by permitting many identity switches. For biologists, HOTA is a more appropriate metric as it better reflects the importance of accurate identity tracking over time. This discrepancy is less evident in existing datasets with shorter tracks, where the likelihood of identity switches is lower.

## 5 Limitations

In this section, first we will highlight the limitations of our dataset and then discuss the shortcomings of the detection and tracking models.

There are three main limitations to our dataset. First, we performed quality checks and improved several manual MOT annotations (see supplementary). Despite these efforts, some bounding box annotations in our dataset do not reflect the exact size of the animals. Tighter or looser bounding boxes were introduced during the pre-labelling stage, or during the MOT annotations with DarkLabel (See Fig 4 for examples). These variations themselves reduce the localisation IoU, in turn affecting the HOTA score. Second, sub-adult males and females are similar in appearance. Hence, the data may include a few misclassified individuals causing errors in detection and classification by the models (Fig 4). However, most trackers do not consider the classification errors (male/female) during tracking assignment and seemingly assign the right identity ID based on the localization of bounding boxes (see Fig 4). Third, in the Re-ID dataset, we do not know the exact number of unique individuals in the dataset. It is possible but unlikely that same individuals were present at same location between different days. We hope to have captured sufficient variation in individuals by annotating videos across different days, in two different regions of the lek. It is for this reason that we limit our Re-ID dataset to identifying individuals within video segments rather than between segments.

Next, we highlight limitations of the state-of-the-art methods in the context of our dataset. Several types of detection and tracking errors occur when individuals are in close proximity. This results in multiple individuals being detected as a single individual, loss of tracking, and identity switches (Fig

---

[4]https://github.com/mikel-brostrom/yolo_tracking
[5]https://github.com/JonathonLuiten/TrackEval

4). This problem is reduced in feature based tracking methods, however, such considerations can improve localization based trackers. As mentioned above, localisation-based trackers do not consider class assignment while associating tracking results. This observation also reveals a limitation of the existing evaluation metrics such as HOTA and MOTA. These metrics do not handle multi-classes instances while evaluating multi-object tracking and therefore our evaluation will not reflect errors due to misclassifications. Evaluation for MOT in this paper is therefore done using a single category i.e., blackbuck.

# 6    Future directions

Tracking and identifying wild animals using UAVs is a promising and rapidly evolving research direction. While our dataset addresses the MOT problem with a large group of animals, it is focused on a single species in a specific behavioural context. To build a more generalisable animal tracking approach, larger datasets that encompass a broader diversity of animals and habitats are essential. Solving the Re-ID problem for animals that look very similar poses a significant challenge. New ground truthing approaches are needed to consistently identify the same individuals across days, seasons, and even years.

Another key challenge is designing computationally efficient methods to process large volumes of high-resolution video while maintaining high accuracy. Our fastest model runs at 32 fps but has significant errors, while the best-performing model operates at 3 fps (see supplementary), which is insufficient for large-scale, long-term UAV-based monitoring programs. One potential solution to resolve this is involves training smaller models by splitting one high-resolution image into multiple images of lower resolution, an approach we will explore in the future.

Beyond new datasets and methods, improved evaluation metrics are necessary to assess the effectiveness of algorithms and dataset complexity in the context of behaviour monitoring. For example, while HOTA is useful for evaluating tracking over long periods and accounting for identity switches, it does not accommodate multiple classes in MOT. Consequently, we simplified the classification of males and females into a single blackbuck class for evaluation. Additionally, more advanced analyses techniques are needed to assess the complexity of datasets within the framework of computer vision challenges. As shown in Sec.4, we use movement analysis to demonstrate that our dataset captures various types of animal movements. Standardized analysis techniques will help identify gaps in datasets and provide a systematic method for capturing a wider range of behavioural patterns.

# 7    Conclusion

BuckTales is first large-scale UAV dataset to address multi-object tracking (MOT) and animal re-Identification (Re-ID) problem for studying wild animals in natural environments. These data were collected with biologists using a data collection scheme using multiple simultaneously-flying UAVs. We claim that our dataset captures the whole range of behaviours exhibited by blackbuck as part of a rare mating strategy and thus it is suitable to address the MOT problem with wild animals. We back these claims with specific movement analyses and benchmarking of existing trackers. BuckTales contributes some of the longest sequences of video annotations with large number of wild animals for MOT and Re-ID problem. While our dataset contains a unique behaviour, we highlight that similar situations occur throughout the animal kingdom where large groups of animals occupy a large area for limited or extended periods. With the rapid adoption of UAVs within the animal behaviour community, we expect that methodologies developed with our dataset will directly impact wildlife monitoring, conservation and research, and pave to way towards the adoption of a fleet of drones for long-term animal tracking in large open ecosystems.

# 8    Acknowledgement

Support for this project is provided by the Max planck society, Deutsche Forschungsgemeinschaft (DFG) under Germany's Excellence Strategy EXC 2117-422037984 and by the Alexander von Humboldt Professorship endowed by the Federal Ministry of Education and Research awarded to Margaret C Crofoot. We thank Sakshi Rao, Mansi Dave, Kuldeep Chouhan, Aporoopa S R, Shreya William, Ritika Chaterjee, Binay Aswal for annotations and Elisabeth Böker for the quirky title idea.

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

# A   Appendix / supplemental material

## A.1   Dataset availability

The dataset can be accessed using the link: Dataset. The existing structure is shared in three different folders. Each of them contain a README file which explains the structure. We have provided datasets in standard format (detection:COCO, MOT: MOT challenge format) which makes it easier for other to load dataset with standard libraries. Our plan is to submit the entire dataset on publicly available platform such as Zenodo. The code-base used for various parts of the dataset will be provided separately through github links, also uploaded via Zenodo having a permanent code base.

## A.2   Limitations of dataset

The figure provided in this section is to support the claim about limitations in the dataset and further explanation of some of the errors in tracking results. We identified some of these error while doing qualitative observation of the results from best performing models.

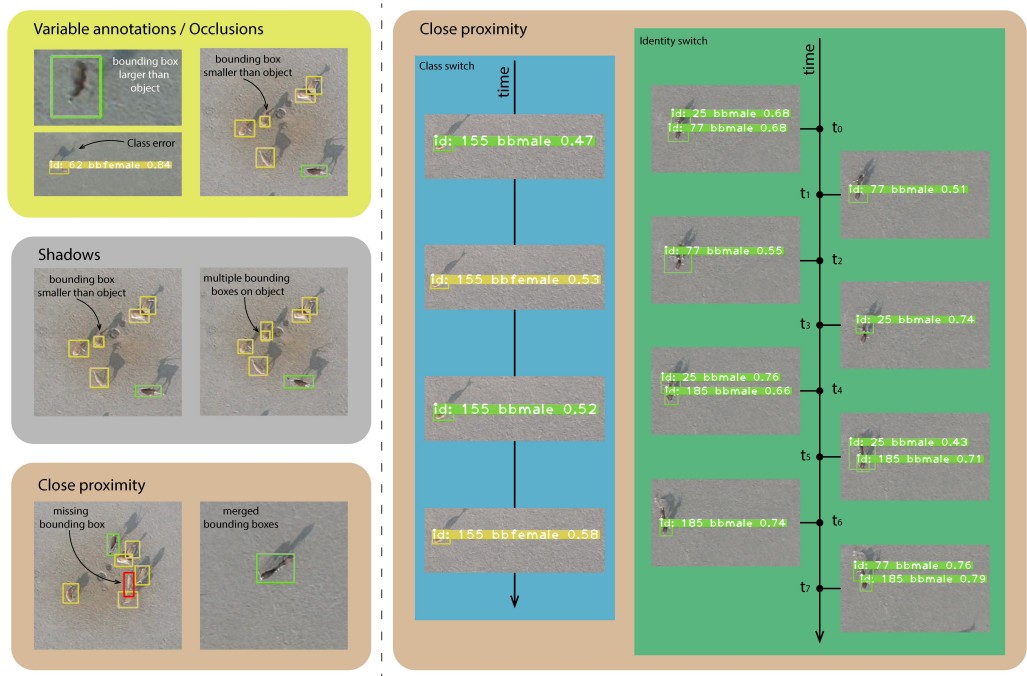

Figure 4: The figure highlighs examples of some limitations of the existing dataset and the state of the art. Images on the left- who that bounding box detection does not always fit perfectly. Shadows and difficult postures can produce missing detections or create confusion with identification of class category. Proximity of the animals also contributes to error in tracking due to missing detection, merged bounding boxes or switching in IDs.

## A.3   Dataset comparison

The table provided in this section covers all the dataset available with UAV based images. There are other datasets with annotations of points for counting animals, however they are not directly compatible with bounding boxes and we have not considered them in this comparison table. Similarly, there are few datasets from other aerial imaging modalities like planes or satellite. We have not included these as well to keep the comparisons concise with drone based imaging.

Table 2: Details of datasets with animals and UAVs

| Object detection problem | | | | | | |
|---|---|---|---|---|---|---|
| Dataset | Categories | Annotations | Type | Images | Pixels | Height (m) |
| [Weinstein et al., 2022] | Multiple | 23765 | RGB | 23765 | 35 | 76-91 |
| [Shao et al., 2020] | Single | 1919 | RGB | 664 | 90 | 50 |
| [Bondi et al., 2020] | Multiple | 166221 | Thermal | 61994 | 35 | 60-120 |
| **BuckTales** | Multiple | 18488 | RGB | 320 | 1600 | 80 |
| MOT problem | | | | | | |
| Dataset | Categories | Tracks | Type | Images | Pixels | Height (m) |
| [Barrios et al., 2024] | Single | 241 | Thermal | 959 | X | Var. |
| **BuckTales** | Multiple | 680 | RGB | 22502 | 1600 | 80 |
| Re-ID problem | | | | | | |
| Dataset | Categories | Instances | Type | Images | Pixels | Height (m) |
| [Desai et al., 2022] | Single | 143 | RGB | 88000 | 1000 | 8-10 |
| **BuckTales** | Multiple | 730 | RGB | 589828 | 1600 | 80 |

## A.4 BoT-SORT with Re-ID networks

We used BoT-SORT as one of the methods which uses not only position but features of the detected targets to perform tracking. There models arguably perform better against the only localization based trackers. All the models used with Re-ID support were trained with human data for Re-ID task and we have not trained them specifically on our dataset. The models are downloaded from torchreid [6] and the selection of the model is based on the performance of the models based on same-domain or cross-domain task. Model *osnet_x0_25* is the best performing model in same-domain Re-ID where as *osnet_x1_0* model is the fastest Re-ID models with same-domain training. *osnet_ain_x1_0* is one of the best perofmring model on the cross-domain Re-ID task.

---

[6]https://kaiyangzhou.github.io/deep-person-reid/MODEL_ZOO.html

