# Supplementary Material

The material provided in this document contains additional information relevant to the dataset. The authors have provided extra details about data collection, annotation clean-up pipeline and evaluation. Finally, we have also provided a datasheet for dataset.

## 1 Dataset details

In this section, we provide additional details regarding the dataset collection process. The link to access the dataset Bucktales. A guide to use the DarkLabel annotation tool is provided with dataset on Edmond. It can also be searched on Edmond platform of the Max Planck Group. The link to annotation analysis code is here.

### 1.1 Data collection

The video recording was done at sunrise and sunset every day between 2-18 March 2023, in Tal Chhapar Wildlife Sanctuary, India. The images for the object detection dataset are selected from nine different days from this period. Peak activity on the lek occurred between 9-15 March 2023. The MOT and Re-ID videos are selected from the peak activity period. All the three drones (DJI Air 2S) were manually controlled and drone positions were kept consistent for every single day. The videos and images offered as part of the dataset are selected based on the notes from the lek activity log prepared in conjunction with the recording sessions. This perspective also allows leveraging existing algorithms and methods produced in the lab environments to study movement of captive animals i.e. insects, birds, or fish. This project is the first ever use of UAVs for studying lekking behaviour.

**Drone relay technique** is a method of using two or more drones in sequential manner by swapping them such that a particular area is monitored continuously for a long time without being affected by the battery life of a single drone. Once the first drone is airborne, the second drone is prepared for take off. After approximately 15 mins of recording, we fly the second drone and position it 15 meters above the first drone. The pilots synchronize with each other to record 5 to 10 seconds of data simultaneously. The first drone is then called back and the second drone replaces the first drone at 80 meter altitude. This technique allows us to record up to 2 hrs of data with eight relay cycles.

Our team consists of six drone pilots, three pilots fly the recording drone and three other co-pilots prepare second set of drones for relay.

## 1.2 Data format

We provide complete dataset in multiple formats. Annotations for object detection are in COCO and YOLO format, whereas MOT problems is provided in MOT challenge format. The Re-ID dataset is provided in a custom format. Each folder in the dataset is provided with a README file. The final project website will also contain all relevant information.

## 1.3 Detection dataset

The dataset is prepared in two stages, in the first stage a small test dataset was prepared using preliminary videos from 2022 (60 images). We trained a FAST-RCNN network to train a generic blackbuck detection model. The generic detector is used to label blackbucks in 320 images from recordings of 2023. A field expert has manually verified all the pre-labelled images, and further added missing annotations with assignment of classes to the detection results. The final dataset contains 24k bounding boxes with 6 different class categories (see table 3). For the multi-object tracking (MOT) task in the paper, we have only focused on the male and female classes. The annotations of other classes such as "dogs" and "birds" are relevant for the behaviour study but do not hold significance in the context of the MOT or Re-Identification dataset presented in the paper.

## 1.4 Re-ID Dataset

The Re-ID dataset contains six sets consisting of 11 pair of videos. Each set consists of annotations of tracks from a specific region, this region is visible in both the videos. Both videos can be stacked vertically on top of each other. We have provided videos augmented with the annotations for easy visualizations. Each video pair contains different number of individuals and different lengths of tracks because annotations are only provided for the animals that are visible in both images at the same time. The details of the annotations for each video pair is given in the table 1.

## 1.5 Behaviour diversity

We provide a list of typical behaviour patterns of males, and females and their interactions at the lek during mating season.

Display walk (M): Male walking by moving legs stiffly with ears held down, tail held up, or in a rising position. The male faces the female and walks of few steps towards her with swift action. It is a repetitive movement with a slight readjustment of position before each display.

| ID | Video_1 | Video_2 | Tracks | Track length | | |
|---|---|---|---|---|---|---|
| | | | | shortest | average | longest |
| Set 1 | V1_0312 | V2_0575 | 62 | 8 | 936.03 | 5811 |
| Set 2 | V1_0294 | V2_0310 | 129 | 5 | 672.49 | 5793 |
| Set 2 | V1_0310 | V2_0573 | 66 | 20 | 905.19 | 5798 |
| Set 3 | V1_0020 | V2_0940 | 109 | 10 | 724.86 | 5220 |
| Set 3 | V1_0940 | V2_0923 | 65 | 7 | 616.96 | 4542 |
| Set 4 | V1_0079 | V2_0001 | 53 | 20 | 1022.90 | 4726 |
| Set 4 | V1_0001 | V2_0987 | 46 | 20 | 609.15 | 5292 |
| Set 5 | V1_0080 | V2_0002 | 53 | 7 | 1040.49 | 5136 |
| Set 5 | V1_0002 | V2_0988 | 40 | 63 | 726.92 | 5306 |
| Set 6 | V1_0081 | V2_0003 | 74 | 2 | 858.35 | 5137 |
| Set 6 | V1_0003 | V2_0989 | 33 | 20 | 1161.72 | 5300 |

Table 1: Details of video pairs included in the Re-ID dataset along with an overview of the track length.

Courtship (M-F): Male following a female in close courtship walk with ears held down, tail held up, and nose-up display. The behaviour can take place for a few to several minutes. It is repetitive in terms of movement.

Scent-marking (M): Male holding the pose of urinating or defecating.

Fight (M-M): Males engaged in escalated horn clash with intermittent locking and disengaging of horns. Males go back and forth for locking horns, the fight may last a few seconds or minutes.

*Parallel walk (M-M): Two males walk side by side for several seconds, supposedly to size up each other and decide if they want to engage in fights.

*Chase (M - M): The males might chase each other away from their territories, this is very dynamic. Typically, sub-adult males coming on the lek are chased away by separate males. Chase is generally short, a few seconds.

Walking (M): Male walking with ears and tail in normal position.

Lying (M): Male lying on dung pile with ears in normal position. Duration depends on the individual.

Walking (F): Female walking and moving between territories.

Sitting (F): Females sitting on the territories when disinterested in copulation. Duration depends on the individual, the presence of females on the lek, and other factors such as time of day.

Mounting (M-F): The male trying to mount the female from behind. The duration of this behaviour depends on females. Mounting attempts can last a few seconds to minutes.

## 2  MOT Annotations with DarkLabel

In this section, we will offer details regarding the use of DarkLabel tool for annotation. Further, we will describe errors that may be introduced during the

Table 2: Per-video distribution of annotations in the MOT dataset

| Videos | Frames | Duration (sec) | Annotations | Tracks | Average animals |
|--------|--------|----------------|-------------|--------|-----------------|
| DJI_0190 | 5805 | 193.6 | 399310 | 127 | 68 |
| DJI_0771 | 3928 | 131.03 | 233199 | 87 | 59 |
| DJI_0158 | 3101 | 103.44 | 193357 | 67 | 62 |
| DJI_0312_2 | 1651 | 55.06 | 113295 | 86 | 68 |
| DJI_0782 | 1502 | 50.08 | 67703 | 52 | 45 |
| DJI_0744 | 1501 | 50.05 | 35660 | 31 | 23 |
| DJI_0766_2 | 1213 | 40.44 | 45229 | 54 | 37 |
| DJI_0312_1 | 802 | 26.73 | 51317 | 74 | 63 |
| DJI_0175 | 752 | 25.06 | 31334 | 44 | 41 |
| DJI_0176_1 | 489 | 16.28 | 11120 | 24 | 22 |
| DJI_0766_1 | 445 | 14.81 | 16078 | 41 | 36 |
| DJI_0176_2 | 315 | 10.48 | 7560 | 24 | 24 |
| Total | 21504 | 717.12 | 1205162 | 711 | |

annotation process and offer some ideas for designing sanity checks that can be used to identify and correct the errors in annotation.

## 2.1 Annotation tutorial

We have made a short video showing the entire process of annotation. This guide is made for colleagues from biology. The link can be accessed with hyperlink here.

## 2.2 Annotation error

The MOT annotations are performed under supervision of the experts. A team of four members undertook the job of annotating each frame using DarkLabel. The complexity of the task is high because each frame consists of a large number of individuals and new individuals feature dynamically. The team used the annotation propagation feature of the software. We learned that this feature can often introduce some error in the annotation which may have to be corrected later. The following text provides details of these errors and measures taken to correct them. The code used for correction of the annotations is uploaded here.

### 2.2.1 Duplicate Bounding Box Error

In certain cases, we encountered a single frame containing two bounding boxes with identical IDs. These duplicate boxes may either overlap each other or be assigned to different individuals within the frame. To identify such instances, we wrote a script that performs a check to locate any unique IDs that appear

more than once in a given frame. The code then reports the ID and frame number where the duplication occurs in the output text file.

We wrote a separate script that can correct some of these errors automatically. When the code detects two bounding boxes with the same ID within a single frame, it compares these duplicates with the bounding box of the corresponding individual (using ID) from the previous frame. The code then calculates the distance between the two duplicate bounding boxes in the current frame. If the distance between the duplicates exceeds 100 pixels, the code computes the distance between each duplicate and the coordinates of the same individual in the previous frame. The duplicate bounding box with the larger distance from the previous frame is then removed. On the other hand, if the distance between the duplicates is less than 100 pixels, the code compares the area of each duplicate bounding box with the area of the same individual's bounding box from the previous frame. The duplicate that exhibits the smallest difference in area compared to the previous frame is retained, while the other duplicate is removed.

### 2.2.2   Class ID error

In some rare instances, the class ID value in certain frames may change to *-1*, which is not assigned to any category of interest. When this error occurs, the affected bounding boxes in the frame are displayed with numbers instead of the expected category labels. No particular cause is known for the occurrence of this error but our observation suggests that the error typically affects all individuals within the specific frame and only affects class assignment but not the tracking annotation. To identify this error, we designed a script to identify individuals with class ID as *-1* and generate a report with the ID and their frame number.

We wrote a script to automatically correct this error in some cases using information from the annotation in the previous frame (see figure 1). We calculate the Intersection over Union (IoU) between all the faulty bounding box and every bounding box present in the previous frame. The bounding box from the previous frame that yields the largest IoU value is selected as the best match for the faulty bounding box. This selection is based on the assumption that the individual with the highest IoU in the previous frame is the most probable candidate for the current faulty bounding box. The class assignment for the faulty bounding box is changed accordingly.

### 2.2.3   Disappearing ID error

This error occurs when an individual's ID number disappears from the tracking data and then reappears in subsequent frames. The disappearance of an ID can lead to unwanted gaps in trajectories and possibly affect the evaluation for the continuous tracking of that individual throughout the video.

Our script compares the presence of each individual ID in a particular frame (n) with their presence in the previous frame (n-1). If an individual is missing, the code generates a summary report in a text file specifying the missing

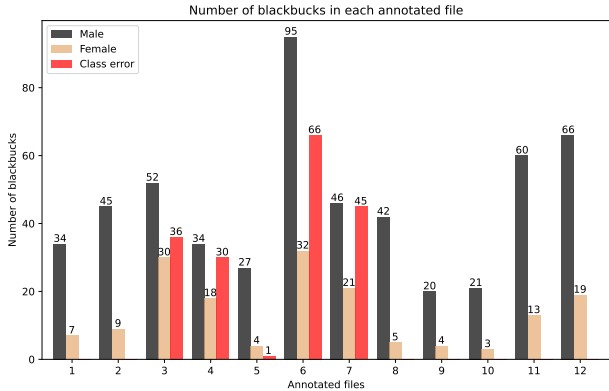

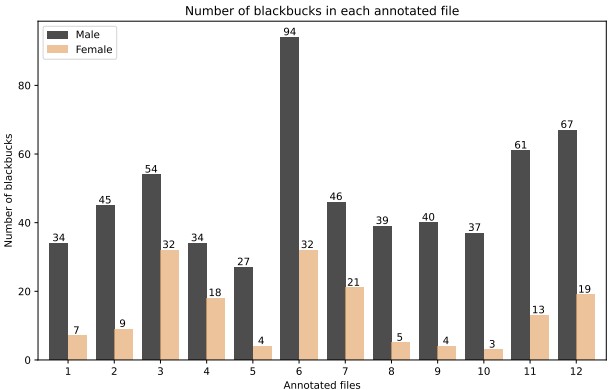

Figure 1: The figure shows class ID errors before and after the correction process. The red bars in the top figure shows error.

individual along with the frame number where the ID went missing.

The disappearing ID error can only be resolved through manual correction using the output text file as a reference. The annotator can utilize the DarkLabel software to navigate to the specific frame where the error occurs, based on the information provided in the text file. Once at the correct frame, the annotator can manually annotate the missing bounding box for the individual whose ID disappeared. By manually adding the missing annotation, the annotator ensures that the individual's tracking information remains consistent throughout the video, mitigating any issues caused by the temporary disappearance of the ID.

### 2.2.4   Bounding box jump

We observed that bounding box in some cases can suddenly move a considerable distance (jump), which is unlikely for a blackbuck under normal circumstances.

This sudden shift happens if the blackbuck starts running or walking suddenly after being stagnant for a while. The automated annotation propagation fails to do continuous detection and abruptly catches up with the individual's new position after a few frames. This indicates the presence of frames where the bounding boxes do not contain any individuals. This error is checked with Intersection over Union (IoU) overlap of an individual's position in subsequent frames. The detection code identifies and reports the ID and frame numbers for each individual that does not show any overlap with IoU in subsequent frames.

To resolve this error, manual intervention is necessary, using the summary report prepared from our script. The annotator can navigate to the specified frame using the DarkLabel software and move backward through the frames until they reach a frame where the individual is correctly positioned inside the bounding box. Once the correct starting frame is identified, the annotator should proceed forward through the frames with automated annotation propagation option while manually adjusting the bounding box position to ensure that the individual remains inside the box. This manual adjustment should continue until the bounding box successfully passes the frame where the error was initially detected. By manually correcting the bounding box position in the frames leading up to and beyond the error frame, the annotator effectively bridges the gap caused by the sudden shift in the bounding box.

### 2.2.5 Area Error

This error occurs when the bounding box area of an individual increases beyond the expected size. Several factors can contribute to this issue, such as errors made by the annotator, rapid movement of the individual, or the bounding box inadvertently including the individual's shadow.

To identify instances of this error, a threshold value is established and applied to all bounding boxes. The generated text file will provide a report indicating the number of frames containing bounding boxes with areas exceeding the threshold, as well as the count of individuals exhibiting boxes above the threshold.

In addition to the text file, graphs are created to visualize how the area of specific individual changes across different frames. These graphs are generated exclusively for individuals whose area has surpassed the threshold value at any point during the annotation process. The graphs serve as a valuable reference tool for correcting the area manually.

To correct this area error, the bounding boxes must be manually edited. The annotator can utilize the generated graphs as a guide to identify the specific frames where the error occurs. By employing the DarkLabel software, the annotator can navigate to the relevant frames and make necessary adjustments to the bounding boxes. During the manual editing process, the annotator should focus on resizing the bounding boxes to ensure that they accurately encompass the individual without including excessive background or shadows. The graphs provide a clear indication of the frames requiring attention, allowing the annotator to efficiently target and correct the problematic bounding boxes. By manually refining the bounding boxes based on the insights provided by the

graphs, the annotator can significantly improve the accuracy and consistency of the annotation, ultimately enhancing the quality of the tracking data.

# 3   Detection experiments

## 3.1   Data split

In this section, we provide all relevant details regarding the training of detection model. To split the detection dataset for model training, we followed the ratio of 0.7:0.15:0.15 for train, validation, and test set (see table 3). Due to unbalanced class distribution among objects and across frames, it is not feasible to make every class comply with the expected ratio in the split. With our best effort to approximate the ideal ratio, our split statistics is shown in supplementary. The same training, test, and validation sets are provided with the paper.

We observed that detection datasets includes some rare categories such as dogs (in our dataset are mostly annotated as unknown) and birds, inclusion of these categories during the training stage reduces the accuracy probably due to low representation in the dataset. Detection of rare categories is a challenge at the moment, however we made sure that videos provided for MOT evaluation does not contain other animals and therefore recommend training detection models only with annotations provided with males and females.

Table 3: Image distribution in the detection dataset with all categories.

| Class | Total (320 images) | Train (219 images) | Val (50 images) | Test (51 images) |
|---|---|---|---|---|
| bbfemale | 4481 | 3119 | 786 | 576 |
| bbmale | 14002 | 9625 | 2249 | 2128 |
| shadow | 2388 | 1585 | 379 | 424 |
| drone | 50 | 34 | 8 | 8 |
| bird | 70 | 47 | 10 | 13 |
| unknown | 102 | 70 | 15 | 17 |

## 3.2   Detection experiments

We evaluated different detection models in two phases to select a best-performing model for tracking evaluation. We also conducted a basic investigation to find the influence of different factors on the detector's performance. In the first phase, we compared the performance of YOLOv8n, YOLOv8m, YOLOv8x, and Detectron2 with different image sizes. Table 4 shows the mean Average Precision (mAP) scores of Blackbucks ("bb", averaged by mAP of "bbmale" and "bbfemale") for each model and image size combination. Each model are trained with only "bbmale" and "bbfemale" class. We found that using the original im-

age size (5472) yielded the best performance across all models, and YOLOv8 performs better than Detectron 2 with all model sizes.

Table 4: mAP of different models with different image sizes

| Model | Image Size | mAP |
|---|---|---|
| YOLOv8n | 1280 | 0.4696 |
| YOLOv8m | 1280 | 0.5209 |
| YOLOv8x | 1280 | 0.5297 |
| YOLOv8n | 2560 | 0.6081 |
| YOLOv8m | 2560 | 0.6415 |
| YOLOv8x | 2560 | 0.6448 |
| YOLOv8n | 5472 | 0.6659 |
| YOLOv8m | 5472 | 0.6801 |
| YOLOv8x | 5472 | 0.6730 |
| Detectron2 | 5472 | 0.5748 |

In the second phase, we focused on the influence of class setups on the detector's performance. We trained YOLOv8m and YOLOv8x models with different class configurations using the original image size (5472). Table 6 presents the mAP scores for each class under different class setups, but we primarily focus on mAP of "bb". The results show that the detector's performance remains relatively consistent across different class setups. However, the inclusion of additional classes such as shadow, drone, bird, and unknown slightly decreased the overall performance on Blackbucks.

The models in the second phase were trained with NVIDIA A100 (80GB) GPUs with an early stop of 1000 epochs. YOLOv8 keeps the best model with the highest fitness score, which is calculated by $0.1 \times mAP@50 + 0.9 \times mAP_{0.05:0.95}$.

Table 5: Class configurations used for blackbuck detection.

| Class configuration | Classes |
|---|---|
| gd (gender) | bbmale, bbfemale |
| gd_shadow | bbmale, bbfemale, shadow |
| gd_drone_bird | bbmale, bbfemale, drone, birds |
| rm_unknown | bbmale, bbfemale, shadow, drone, birds |
| mc (multi-class) | all available classes |
| sc (single-class) | blackbuck (bbmale, bbfemale as combined) |

Table 6: mAP of Classes with YOLOv8 Models Trained on Different Class Setups

| Class Setup | Model | bb | bbmale | bbfemale | shadow | drone | bird | unknown | Best Epoch |
|---|---|---|---|---|---|---|---|---|---|
| gd | YOLOv8m | 0.6259 | 0.6486 | 0.6032 | - | - | - | - | 529 |
| gd | YOLOv8x | 0.6220 | 0.6392 | 0.6047 | - | - | - | - | 405 |
| gd_shadow | YOLOv8m | 0.6248 | 0.6455 | 0.6040 | 0.2495 | - | - | - | 104 |
| gd_shadow | YOLOv8x | 0.6192 | 0.6428 | 0.5957 | 0.2475 | - | - | - | 125 |
| gd_drone_bird | YOLOv8m | 0.6089 | 0.6300 | 0.5874 | - | 0.3788 | 0.4266 | - | 90 |
| gd_drone_bird | YOLOv8x | 0.6199 | 0.643d0 | 0.5965 | - | 0.2840 | 0.5031 | - | 325 |
| rm_unknown | YOLOv8m | 0.6167 | 0.6426 | 0.5907 | 0.2548 | 0.3030 | 0.4212 | - | 99 |
| rm_unknown | YOLOv8x | 0.6120 | 0.6320 | 0.5910 | 0.2317 | 0.4135 | 0.3692 | - | 106 |
| mc | YOLOv8m | 0.6110 | 0.6419 | 0.5800 | 0.2466 | 0.4380 | 0.3766 | 0.5139 | 75 |
| mc | YOLOv8x | 0.6027 | 0.6253 | 0.5802 | 0.2311 | 0.3314 | 0.4845 | 0.5385 | 125 |

# 4 MOT analysis & experiments

This section contains additional information about the MOT data and evaluation of the tracking performance. We include some details regarding use of annotations for doing the movement analysis provided in the main text. Furthermore, we also provide additional details of experiments done to assess speed of the trackers. This is practically relevant for biologists to understand that large scale processing with larger models has downside in terms of time.

## 4.1 Smoothing

The raw data exhibited noise in the form of small, frame-to-frame movements of the bounding boxes. To mitigate this noise and highlight the underlying trends, a moving average smoothing technique was employed. Moving average smoothing is a method used to reduce random fluctuations in time series or sequential data by calculating the average value within a sliding window of adjacent data points. In our case, smoothing was applied to the X and Y coordinates of the bounding box center points. Initially, we calculated the center of each bounding box, followed by performing moving average smoothing on the X and Y coordinates of these center points. A window length of 15 was selected for this smoothing process. The choice of a 15-frame window size for the moving average smoothing process was determined based on empirical evaluation, aiming to strike a balance between noise reduction and preservation of true signal patterns. Thus all calculations regarding distance measurements were performed using the smoothed X and Y coordinates.

## 4.2 Distance analysis

In this subsection, we want to inform the readers about distances travelled by blackbucks in different videos. The information given in figure 2 is summarised in the data analysis section of the main text.

## 4.3 Tracking speed evaluation

We used the scripts for MOT16 challenge offered by TrackEval [1] to evaluate trackers. We evaluated the tracking speed of different trackers on our dataset using the default configuration provided by BoxMOT [2] . When Re-ID model is involved, we used a single GPU (NVIDIA Quadro RTX 6000). The speed was measured in frames per second (FPS) and averaged over each sequence. Table 7 shows the tracking speed for each tracker, with and without the use of a Re-ID model.

It is important to note that the tracking speed can potentially be improved by optimizing image IO, as we are currently reading images directly from the video and feeding them into the Re-ID model. Future work could explore more

---

[1] https://github.com/JonathonLuiten/TrackEval
[2] https://github.com/mikel-brostrom/yolo_tracking

efficient methods for image processing and data transfer to enhance the overall tracking performance.

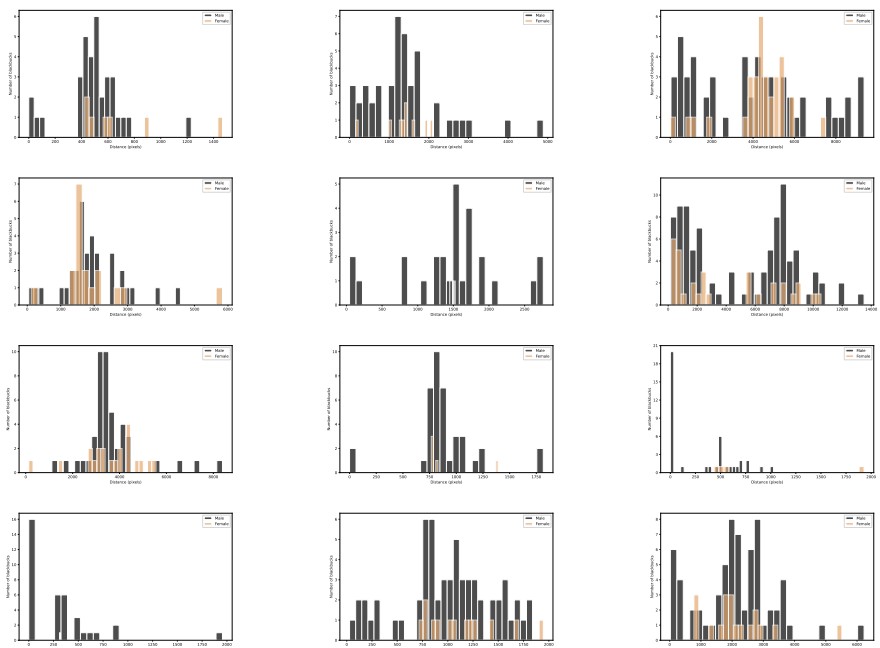

Figure 2: Distance analysis of blackbuck movement across different videos. The image displays histogram with x-axis representing the distance travelled and y-axis representing number of individuals. The males are depicted in black and the females in yellow.

Table 7: Tracking Speed of Different Trackers

| Method | Re-ID | FPS |
|---|---|---|
| BYTE | - | 34.05 |
| OCST | - | 32.8 |
| BoT-SORT | - | 4.19 |
| | osnet_x0_25_msmt17 | 3.41 |
| | osnet_x1_0_msmt17 | 2.88 |
| | osnet_ain_x1_0_msmt17 | 3.28 |
| | lmbn_n_duke | 2.96 |

# 5 Datasheet for dataset

1. Motivation

   (a) For what purpose was the dataset created?
   This dataset was created to facilitate the study of animal behaviour at large scale using UAVs in their natural habitat. Main focus for this dataset is to solve MOT and Re-ID problem (identifying same individual in two video sequences).

   (b) Who created the dataset and on behalf of which entity?
   The details will be added after successful acceptance of the article.

   (c) Who funded the creation of the dataset?
   The details will be added after successful acceptance of the article.

   (d) Any other Comments?
   No.

2. Composition

   (a) What do the instances that comprise the dataset represent?
   The dataset is provided in form of video sequences. Each instance contains aerial video footages of a group of blackbuck (males and females) on the lekking arena (traditional mating ground) at a protected area in India. Annotations mainly include bounding box and classification of all animals in all frames of the sequence provided with the dataset. The dataset is divided into three main parts. MOT dataset consists of videos. Re-ID dataset consists of video pair, which means that each annotation caters to two separate video files. The third type of dataset is a special type of dataset consisting of images for object detection problem. This dataset is specifically provided to solve the MOT problem.

   (b) How many instances are there in total?
   We have 12 video instances in total for the MOT problem and 11 pair of videos for Re-ID problem. Details are provided in Section 4.

   (c) Does the dataset contain all possible instances or is it a sample (not necessarily random) of instances from a larger set?
   Annotated dataset is a subset of 50 hours worth of data recorded in 2023. The video data consists of behavioral activities of antelopes during their mating season. We show analysis of movement and behavior of animals in Section 4 to highlight relevance of dataset with the MOT and Re-ID problem.

   (d) What data does each instance consist of?
   See section 3 for all the details.

   (e) Is there a label or target associated with each instance?
   Yes. All images or frames of provided video sequences are annotated.

(f) Is any information missing from individual instances?
No.

(g) Are relationships between individual instances made explicit?
Yes.

(h) Are there recommended data splits?
We have recommended training, validation, and test datasets for the detection dataset. The MOT and Re-ID annotations are not provided with any train, test or validation set.

(i) Are there any errors, sources of noise, or redundancies in the dataset?
There are almost certainly some errors in video annotations. The limitations are described in section 5 of the paper. Additionally, supplementary material contains detailed account of errors found in the annotations and methods for correcting these errors.

(j) Is the dataset self-contained, or does it link to or otherwise rely on external resources (e.g., websites, tweets, other datasets)?
The dataset is self-contained.

(k) Does the dataset contain data that might be considered confidential (e.g., data that is protected by legal privilege or by doctor-patient confidentiality, data that includes the content of individuals' non-public communications)?
No.

(l) Does the dataset contain data that, if viewed directly, might be offensive, insulting, threatening, or might otherwise cause anxiety?
No.

(m) Does the dataset relate to people?
No.

(n) Does the dataset identify any subpopulations (e.g., by age, gender)?
No.

(o) Is it possible to identify individuals (i.e., one or more natural persons), either directly or indirectly (i.e., in combination with other data) from the dataset?
Not applicable. Our dataset only contains wild animals.

(p) Does the dataset contain data that might be considered sensitive in any way e.g., data that reveals racial or ethnic origins, sexual orientations, religious beliefs, political opinions or union memberships, or locations; financial or health data; biometric or genetic data; forms of government identification, such as social security numbers; criminal history)?
No.

(q) Any other comments? None.

3. Collection process

(a) How was the data associated with each instance acquired?
The collection process is described in section 3.

(b) What mechanisms or procedures were used to collect the data (e.g., hardware apparatus or sensor, manual human curation, software program, software API)?
We used DJI Air 2S drones to collect the aerial footage. Section 3 and supplementary material contains all details regarding collection of the data.

(c) If the dataset is a sample from a larger set, what was the sampling strategy (e.g., deterministic, probabilistic with specific sampling probabilities)?
We recorded activity of animals everyday during the mating season. We used these field notes to identify days of high activity. Then we selected the detection dataset using this activity log prepared in the field. The videos are also selected on basis of manual observation of the peak lekking time.

(d) Who was involved in the data collection process (e.g., students, crowdworkers, contractors) and how were they compensated (e.g., how much were crowdworkers paid)?
The video data was collected by the project leaders and lead authors of this article. The annotations were performed and supervised with a large team of 11 field assistants and three leaders. The compensation for tha annotations was included in the stipend and travel allowance provided during the fieldwork.

(e) Over what timeframe was the data collected?
The data provided in this dataset is collected in March 2023, and it was labelled in 2023-2024.

(f) Were any ethical review processes conducted (e.g., by an institutional review board)?
Not applicable. Our dataset raises no ethical concerns regarding the privacy information of human subjects, as it solely focuses on non-invasive observations of antelope behaviour. However, necessary permission were taken from the forest department of India.

(g) Does the dataset relate to people?
No.

(h) Has an analysis of the potential impact of the dataset and its use on data subjects (e.g., a data protection impact analysis) been conducted?
The use of the dataset will benefit the subject of the study as the dataset would lead to conservation or research efforts. We do not see applications of this dataset which may lead to negative impact on the animal.

4. Preprocessing, Cleaning and Labelling

(a) Was any preprocessing/cleaning/labeling of the data done (e.g., discretization or bucketing, tokenization, part-of-speech tagging, SIFT feature extraction, removal of instances, processing of missing values)?

We have used multiple methods for cleaning the annotations. Detection dataset and the Re-ID dataset is manually verified by multiple annotators and field experts. MOT dataset required specific effort for cleaning. We have provided all the details in the supplementary Section 2.

(b) Was the "raw" data saved in addition to the preprocessed/cleaned/labeled data (e.g., to support unanticipated future uses)?

Yes, we maintain raw data on our institute server.

(c) Is the software used to preprocess/clean/label the instances available?

Yes. The annotation softwares used are Labelbox and section 2 of supplementary provides all the details. We have provided a copy with the dataset for download.

5. Uses

(a) Has the dataset been used for any tasks already?

No, the dataset is prepared by us for our ongoing research on the mating system of the antelopes.

(b) Is there a repository that links to any or all papers or systems that use the dataset?

No, at present the data is being used to study the behavior.

(c) What (other) tasks could the dataset be used for?

This dataset is specifically customised for MOT and Re-ID problems. We provide videos for using temporal constraints. However, one could solve more challenging problems like efficient tracking or one short identification.

(d) Is there anything about the composition of the dataset or the way it was collected and preprocessed/cleaned/labeled that might impact future uses?

The dataset at present is focused on the MOT and Re-ID problem. Although, we have provided the detection dataset, it can not be used outside the context of the provided MOT or Re-ID problem becuase the detection dataset is not diverse enough for building a generic detector.

(e) Are there tasks for which the dataset should not be used?

The usage of this dataset should be limited to developing the MOT or Re-ID solutions for blackbucks and to extend the approach to other species. The authors strongly recommend that the methods developed with this dataset should be used .

6. Distribution

(a) Will the dataset be distributed to third parties outside of the entity (e.g., company, institution, organization) on behalf of which the dataset was created?
Yes, the dataset will be made publicly available.

(b) How will the dataset be distributed (e.g., tarball on website, API, GitHub)?
The dataset will be uploaded on a data sharing repository such as Zenodo or Edmond, all the code will be provided with GitHub.

(c) When will the dataset be distributed?
The dataset will be released to the public upon acceptance of this paper. We have provided dataset privately for review purpose with this link. The data is directly related to unpublished biological study of behavior and therefore authors do not want to reveal dataset publicly before acceptance of the manuscript.

(d) Will the dataset be distributed under a copyright or other intellectual property (IP) license, and/or under applicable terms of use (ToU)?
We release our benchmark under CC BY-NC 4.0

(e) Have any third parties imposed IP-based or other restrictions on the data associated with the instances?
No.

(f) Do any export controls or other regulatory restrictions apply to the dataset or to individual instances?
The data is collected with permission from Indian forest department. The data is not to be used for any commercial activity or for using the dataset outside educational/research use. We also do not recommend use for documentaries or other equivalent uses before consultation with the authors.

7. Maintenance

(a) Who is supporting/hosting/maintaining the dataset?
The details will be provided upon publication. The lead authors of the article will maintain the dataset and improve it in the coming years as the dataset grows.

(b) How can the owner/curator/manager of the dataset be contacted?
The curators can be contacted via the email provided with the dataset.

(c) Is there an erratum?
Currently, no. As errors are encountered, future versions of the dataset may be released.

(d) Will the dataset be updated (e.g., to correct labeling errors, add new instances, delete instances')?
Yes. We aim to add behavioral annotations.

(e) If the dataset relates to people, are there applicable limits on the retention of the data associated with the instances (e.g., were individuals in question told that their data would be retained for a fixed

period of time and then deleted)?

Not applicable. The dataset does not relate to people.

(f) Will older versions of the dataset continue to be supported/hosted/maintained?

Yes, older versions of the benchmark will be maintained on our website.

(g) If others want to extend/augment/build on/contribute to the dataset, is there a mechanism for them to do so?

Yes, please contact us by email.