# OpenReview forum: "BuckTales: A multi-UAV dataset for multi-object tracking and re-identification of wild antelopes"
_NeurIPS.cc/2024/Datasets_and_Benchmarks_Track — NeurIPS 2024 Track Datasets and Benchmarks Poster_

### Official Review · Reviewer_GyVH · 2024-07-22
**In-depth description of the proposed dataset**

**Rating:** 7
**Confidence:** 4

**Review:**

This paper provides an in-depth description of the proposed dataset. The authors detail the dataset properties, including animal types distribution, animal movement under various behaviours, etc. Additionally, the paper thoroughly explains the data acquisition scheme and the annotation methodology, ensuring a clear understanding of the dataset's construction.

The datasets are organized as several remarkable baselines input format, which facilitated the reproduction.

I would suggest including some recent baseline methods of animal detection from well-known conferences or journals in the reproduction experiments of the paper. Since I found that both Korger et al. 2023 and Rathore et al. 2023b used CNN for wildlife object detection, I believed Fast-RCNN and YOLOv8 would be fine. If there are any object detection methods specifically modified for wild animal detection, you could add those as baselines as well.

The dataset is quite unbalanced for objection detection tasks, which should be taken into account when training and evaluating baseline methods. The performance of YOLOv8, trained with different class setups and evaluated across different object types, is presented in Table 6 at supplementary material. However, directly training a model on an imbalanced dataset and testing its performance with a very small number of labels may lead to unreliable accuracy. If you want to train models across different class setups, you could employ strategies like oversampling, data augmentation, and other techniques tailored for imbalanced datasets.

The experiment of models performance with differenct input image sizes is worthy. Nevertheless, I am concerned about the adequacy of object detection for smaller targets. Specifically, with a blackbuck approximately sized at 8x3 pixels when image resized to 1280 pixels, was the model trained on these resized images? Is an object size of 8x3 pixels sufficient for detection using YOLOv8? Additionally, have experiments been conducted by cropping the image into multiple parts and processing them separately with the model?

**Strengths:**

The paper provide a large scale UAV dataset for MOT and Re-ID in wild animals.

The paper is well-writen. The dataset and experiments are well-organized and detailed introduced.

**Additional Feedback:**

n/a

**Clarity:**

This paper is well written, and provides an in-depth description of the proposed dataset.

Minor typos:

Figure 3 Caption: 'show' and 'shows' should be consistent

Suggestion:

Table 1: Indicate in Bold the best performance of each metric

**Correctness:**

When I open `MOT_Example_Annotated.MP4` with the default Windows video player, I get an error, but I can successfully read each frame of it using opencv-python.

The md5sum of my downloaded mp4 file is `fa9cfebf43542c8aaceef043f98c857c`.

**Documentation:**

This paper is a sufficient description of the proposed dataset. README file are included in the dataset link. Since the dataset is organized in the baseline format, the baseline reproduction code would be the original baseline code.

**Ethics:**

No. The author claims that the permission obtrained from ethics committee for conducting the study has been provided.

**Limitations:**

Yes. The author discusses the limitations of the work in Section 5.

**Opportunities For Improvement:**

The experiments can be optimized following the suggestion of Review part above.

**Relation To Prior Work:**

Yes. The author claim that the proposed dataset is the first large scale UAV dataset to tackle the problem of multi-object tracking and re-identification in wild animals.

**Summary And Contributions:**

This paper proposed the first large scale UAV dataset to tackle the problem of multi-object tracking and re-identification in wild animals. The dataset consists of 1.2 million annotations of 690 tracks across 12 video clips of 5.4K resolution, with videos averaging 66 seconds in length and featuring 30 to 130 individuals per video.

---

> ### Author Rebuttal · Authors · 2024-08-16
>
> Thank you for an encouraging review of our work. We are delighted that you find our work exciting and offer support for acceptance. Below, we address some of your main concerns.
>
> Q1.	Baseline methods for animals
>
> A1.	The use of Fast-RCNN was inspired by Koger et al., and Rathore et al. (a co-author of this article) use a similar method for detection. Additionally, we used YOLOv8 for blackbuck detection which performed reasonably well as a baseline for our dataset. Apart from these, we did not find specialized methods tailored for animal detection.
>
> Q2.	Unbalanced dataset
>
> A2.	The dataset is indeed unbalanced, as is common in natural wild settings. Our focus for the paper is on blackbuck MOT, and we demonstrate that we achieved reasonable detection accuracy for male and female blackbucks. Other categories (e.g., drone, bird) appearing in the detection dataset are rare and less relevant for the blackbuck MOT problem. We even exclude them from the videos MOT dataset. We agree that augmentation experiments with these categories could be useful for future studies with the entire biological dataset. We will clarify this in Section 4.2.1. that experiment with multi-category detection (Table 6) was a preliminary experiment (for the benefit of biologists) to see the impact of rare categories on overall detection and MOT performance. Based on the results we decided to train with limited categories to provide the best possible benchmarking for trackers.
>
> Q3.	Detection with different image sizes
>
> A3.	Yes, we downsampled the images in our experiment. This experiment was designed to assess the speed-accuracy trade-off for training a possible detection model. For our smallest images 1280p, the reviewer is right that object sizes became too small which is reflected in the poorer performance of this model. As indicated by the reviewer, downsampling images reduces the number of pixels per animal (PPA). We can add this information, it may help biologists who may want to collect data in an informed way (e.g. using PPA to decide the recording altitude, drone resolution, etc.). We will include a discussion on this topic in Sec 3 of supplementary with other information on detection experiments.
>
> Q4.	Experiments with cropped images
>
> A4.	This is a great suggestion and we will include it for future work! The crop experiment is likely to enhance performance, especially in terms of speed as we can train smaller models. However, we think this experiment is unlikely to improve the overall accuracy of detection. We think this because we have trained larger YOLOv8 models (m and x size) with full resolution 5.4K and both have similar performance which suggests that maybe we may need more data to improve results with YOLOv8x. Certainly, in the future, we aim to train smaller models like YOLOv8-n with cropped images rather than downsampling. We can add this discussion of results and convey the essence of the performance problem with a new section on future work. (see rebuttal to reviewer 1)
>
> We agree to correct the typos pointed out by the reviewer and we agree to highlight the result from table in bold.
>
> We apologize that the reviewer were not able to open: MOT_Example_Annotated.MP4. We offer to upload a downsampled version with the dataset for viewers to easily access the files. The authors have often observed problems with opening high-resolution files using Windows player. Our students used KM player, and VLC also does not work reliably.
>
> We hope that the offered changes can be accommodated with the additional space offered in the camera-ready version.

---

> > ### Comment · Reviewer_GyVH · 2024-08-19
> > **Reply**
> >
> > Thank you for your response. Just to clarify, my suggestion to train on cropped images was intended to address the low PPA observed when discussing different image sizes, particularly in relation to its impact on mAP. By cropping rather than resizing, the model can be trained on smaller images without sacrificing PPA. I hope this helps in refining your approach.

---

> > > ### Author Rebuttal · Authors · 2024-08-19
> > >
> > > Dear reviewer,
> > > Thanks for the clarification on the comment. We will use this suggestion for future efforts.

---

### Official Review · Reviewer_TLDH · 2024-07-23
**UAV antelope tracking**

**Rating:** 5
**Confidence:** 3
**Clarity:** The paper is readable but the writing…

**Review:**

One weakness is the short length of trajectories, even though the authors claim these are some of the longest annotated sequences available. Is there a technical reason for these incredibly short annotated recordings? It seems like the original dataset is longer and 'contains 30 min to 120 min' long video sequences. From a biological perspective - how long would movies need to be to gain better understanding of lekking behavior? The authors mention that different types of behaviors are captured in each video but this is far from understanding outcomes.
The 're-identification' problem described here is much simpler than that of actually identifying the same animal on different days or in different contexts. While this is clear in the text, the more general problem of animal ID could be further discussed as a future direction.
If the videos are taken at a 90 degree angle, is it possible to calculate actual distance and speed metrics instead of using pixels, which would depend on detector distance and resolution?
Overall, it is unclear what can be done with this data to benchmark and improve potential multi-animal tracking or identification algorithms.

**Strengths:**

This manuscript attempts to address both computer science and biology audiences for tackling an interdisciplinary problem. The problem of a general lack of biologically relevant datasets (particularly of wild animals in complex environments using arial imaging) is well-motivated.

**Additional Feedback:**

NA

**Correctness:**

The claim that this dataset 'captures the whole range of behaviors exhibited by blackbuck' is not supported beyond simple movement analyses.

**Documentation:**

The authors say data will be made publicly available after acceptance

**Limitations:**

The authors could further discuss how this dataset in particular has fairly well-distributed animals whereas other natural applications have very dense animal populations in environments with more occlusions.

**Opportunities For Improvement:**

While the dataset is interesting, the number of samples is limited, as is the type of behavior exhibited. It is unclear how good the 'ground truth' tracks and identities are in this data, as the authors mention that there are swaps and mis-identified individuals. More clarity on these errors, as well as discussion of limitations in the greater context of tracking in the wild would be useful. The hope that the 'methodologies developed with our dataset will directly impact wildlife monitoring..' is vague given that there is little discussion of the types of future work that need to be done to implement such technologies.

**Relation To Prior Work:**

The authors mention some multi-object tracking datasets with wild animals. More could be included on lab-based methods where multi-animal tracking is performed (JABBA, idTracker, markerless bee tracking using CNNs, etc) and the issues with using these methods in the wild could be discussed.

**Summary And Contributions:**

This work describes a dataset of UAV-captured and tracked sessions of wild animals lekking. This work is aimed at extending recording and tracking of animal behavior to the wild, and addresses the problem of multi-animal tracking and, more weakly, the problem of animal re-identification.

---

> ### Author Rebuttal · Authors · 2024-08-16
>
> We thank the reviewers for their valuable comments and we suggest the following changes to improve the manuscript ( please also see reviewer 2 rebuttals due to similarities in questions).
>
> Q1.	Trajectory duration
>
> A1.	Annotating MOT with multiple individuals over thousands of frames is challenging. We’ve curated 1.2M bounding boxes, making this one of the largest UAV-based MOT datasets for animals. These annotations are valuable for extracting a preliminary behavioral repertoire of blackbuck on leks and making initial inferences about territories that are more or less frequently visited by females (giving glimpses into which individuals are likely to pass on their genes to the next generation). While more detailed insights will require longer data, the current dataset is also sufficient for training detection and tracking algorithms, as evidenced by our baseline results. The eventual goal is to develop algorithms that can be applied to our >60 hours of footage to study blackbuck leks more comprehensively. Hence, the "shortness" of the data is relative; while it might not yet provide deep biological insights, it is adequate for algorithm development and gaining preliminary insights.
>
> Another limitation is technical: the DJI Air 2S drones used split recordings (~15 min) into smaller ~3 min videos. Our longest clip is 5804 frames, reflecting this constraint.
>
> Annotation tools also pose a limitation, i.e. high-resolution images become difficult to annotate with existing MOT tools, especially if hundreds of animals are marked. The semi-automated detection is rather slow. We will add these practical limitations with an explanation (Sec 2 of supplementary).
>
> Q2.	Limited behaviors in data / biological relevance
>
> A2.	We agree that capturing behaviors is not the same as understanding outcomes, which is beyond this paper's scope. The goal here is to highlight the diversity of movement-based behaviors in the data, which can be leveraged in future studies to extract biological insights. To clarify, we will add a list of behaviors exhibited by blackbucks at leks (see below), enhancing Fig 4's illustration of behavioral diversity. It should be noted that one of the goals of automated tracking is to identify new behavior patterns and strategies used by individuals over longer time scales. We also direct the reviewer to our response to Reviewer 2, where we discuss the direct relevance of this dataset to biology-computer research.
>
> Q3.	Conversion to metric coordinates
>
> A3.	The reviewer is correct that tracking these animals in pixel coordinates is not very helpful and that we should move to metric coordinates. This is exactly our plan. We will geo-reference our tracks on an orthomosaic that we have generated using methods described in Koger et al. 2023. This is exactly how we plan to link individual trajectories between neighboring drones.
>
> Q4.	Quality of ground truth
>
> A4.	We apologize for any confusion regarding data quality. To clarify, our dataset is error-free to the best of our knowledge. We mentioned potential errors only to acknowledge the possibility of human mistakes during labeling. In fact, we have additionally taken measures to identify and correct common annotation errors (see Sec 2 supplementary).
>
> Other comments,
> We agree to declare in limitations that our dataset does not consist of extremely dense populations with large occlusions. In related work, we excluded lab-based methods because we focus on the UAV-based approaches. We agree to add issues regarding the use of these methods in the wild.
> Regarding correctness, we will clarify that data analysis is done to show movement profiles of individuals and not types of behaviors.
>
> Behaviors list: This includes individual behavior of males, and females and their interactions such as courtship and mating, assessment displays, and fights. Here we explain some of the known behaviors at lek:
>
> Display walk (M): Male walking by moving legs stiffly with ears held down, tail held up, or in a rising position. The male faces the female and walks of few steps towards her with swift action. It is a repetitive movement with a slight readjustment of position before each display.
>
> Courtship (M-F): Male following a female in close courtship walk with ears held down, tail held up, and nose-up display. The behavior can take place for a few to several minutes. It is repetitive in terms of movement.
>
> Scent-marking (M): Male holding the pose of urinating or defecating.
>
> Fight (M-M): Males engaged in escalated horn clash with intermittent locking and disengaging of
> horns. Males go back and forth for locking horns, the fight may last a few seconds or minutes.
>
> *Parallel walk (M-M): Two males walk side by side for several seconds, supposedly to size up each other and decide if they want to engage in fights.
>
> *Chase (M - M): The males might chase each other away from their territories, this is very dynamic. Typically, sub-adult males coming on the lek are chased away by separate males. Chase is generally short, a few seconds.
>
> Walking (M): Male walking with ears and tail in normal position.
>
> Lying (M): Male lying on dung pile with ears in normal position. Duration depends on the individual.
>
> Walking (F): Female walking and moving between territories.
>
> Sitting (F): Females sitting on the territories when disinterested in copulation. Duration depends on the individual, the presence of females on the lek, and other factors such as time of day.
>
> Mounting (M-F): The male trying to mount the female from behind. The duration of this behavior depends on females. Mounting attempts can last a few seconds to minutes.
>
> Reference: “Territorial and mating strategies of males in a lekking
> population of blackbuck (Antilope cervicapra)” by R. Jayabharathy
> *Additional information added from authors.

---

### Official Review · Reviewer_odky · 2024-07-24
**Interesting dataset with some limitations and an unusual definition of Re-ID**

**Rating:** 5
**Confidence:** 4
**Correctness:** Yes.  Constructed in a sound way.
**Clarity:** Yes

**Review:**

I have mixed feelings about this paper.  The dataset is unusual, timely and interesting.  Real data from animal behaviors in the wild that is high-reslution, high-frame rate and extended is very interesting.  However, the integration between biologists and computer science is shallower than it could be… it seems like the blackbuck tracking problem is chosen, but then the tracking problem is considered as a completely vanially tracking problem — without taking the biological needs into account to characterize how good performance needs to be to answer real biological questions.

Finally, the Re-ID aspect of the paper is also interesting, but this is an unusual definition of Re-ID.   The weaknesses (opportunities for improvement below) highlight reasons why I am concerned that this would not be a dataset that is broadly used by the ML community; and it isn’t clear that this dataset is directly useful as is in the bio-community.

**Strengths:**

a. the dataset is unusual in sharing video of large scale, biologically relevant behaviors.
b. The baseline approaches for the MOT problem are reasonable.
c. Collecting the data (with drone relay process and teams of many pilots coordinating) is complicated and having this kind of data is valuable

**Additional Feedback:**

None

**Documentation:**

Yes

**Ethics:**

No concerns

**Limitations:**

Yes

**Opportunities For Improvement:**

a. I think the biggest opportunitiy for improvement would be tighter integration of the dataset, the analysis and the problem domain.  HOTA, DETA, etc… are metrics used in the MOT domain, but it isn’t clear if there is a threshold that makes MOT “good enough” for the biological questions (perhaps there are different definitions of good enough for different questions), or even if the metrics are measuring the right kinds of things.  r example, tracking might work great when animals are moving slowly, and they almost always move slowly, but if the biologically interesting features is what they do when they move fast, and the tracker breaks then

b. Second, clarifying the role of “Re-ID”.  Usually, this is the problem of re-identifying an object (car/person/?) in another camera view at another time, and the question is how to generate features that support matching across variations in viewpoint, pose, (and weather etc…).  This version, of matching objects viewed in two cameras at the same time, is unusual.
 It isn’t clear to me if the point of the Re-ID annotations is to support future work on the problem here (matching images captured simultaneously) or as data to support re-ID for other versions of the problem where the images are not captured simultaneously.

**Relation To Prior Work:**

Yes

**Summary And Contributions:**

This paper describes the collection and plans to share a dataset of drone video capturing blackbuck lekking behavior.  This is representative of a biological problem domain where collective animal behavior needs to be observed over a wide area and over a long time frame, and multiple drones are required to see everything.

The dataset is (by today’s standards) moderately sized, with 12 video clips, 5.4K resolution, a total of 730  animals, and 1.2 million annotations and 680 tracks.

Data is organized with bounding boxes extracted for object detectors, and results are reported for (multi-) object tracking on this dataset.

---

> ### Author Rebuttal · Authors · 2024-08-16
>
> We thank the reviewer for acknowledging our extensive data collection efforts and are pleased they find our work timely and interesting. We address their concerns below. We also request you to refer to the rebuttal to Reviewer 3, due to similarity in some concerns.
>
> Q1.	Biological relevance
>
> A1.	We apologize if the integration between biology and computer science wasn't clear. We believe our dataset demonstrates a deep integration,
>
> A substantial body of biological literature uses movement data to gain insights into animal social interactions and decision-making [Smith et al. 2021, Sridhar et al. 2021,2023, Rosenthal et al. 2015, Sosna et al. 2019]. While this research is often done in controlled lab settings, our dataset makes similar analyses possible in the wild. Although UAVs have made such recordings more common, biologists often rely on manual correction of tracking results post-recording (Rathore et al. 2023, Koger et al. 2023). There is no suitable dataset for benchmarking tracking methods for the wildlife MOT problem. Our dataset bridges this gap by facilitating biologists to extract behavioral insights from animal movements in natural settings (highlighted in Sec 2).
>
> Our paper clarifies that work is conducted in collaboration with biologists and forest officials, and the algorithms developed on this dataset will influence biological insights from >60 hours of blackbuck lek footage. Soon, detection methods will already aid in organizing an automated census of the blackbuck on a lek for forest officials. We will highlight that processing large video datasets is a big challenge for biologists, which we have exposed by showing that current state-of-the-art trackers are slow and suffer from identity switching. We offer the addition of a separate future work section to clear ideas on advancing the state of the art with our dataset (also advised by Rev 1).
>
> Blackbuck, being moderately sized and among the fastest terrestrial mammals (capable of running up to 80 km/h), presents a broad range of tracking challenges relevant to animal behavior studies. Effective tracking of blackbuck could generalize to other similarly sized or larger mammals. We show with movement analysis that our dataset includes a variety of interactions at different speeds and densities, providing a solid ground truth for tracking animals over relatively long trajectories.
>
> We appreciate the reviewer’s point on defining “good tracking” in biological contexts. We have a short discussion on this point in Sec 2.1, which can be clarified with further additions of examples. We agree that the definition of good tracking can vary depending on the biological questions at hand. Here, we define the problem as a pure tracking problem because the ideal goal is to fuse the trajectories of all individuals across the entire recording session as they move from one drone view to another.
>
> The reviewer also raised questions on the selection of correct evaluation metrics for tracking wildlife. Currently, there is no consensus or opinion on this point, thus we used commonly known metrics. In line with this, we will add the following text in section 4.2.2: “Our dataset highlights a known limitation of MOTA, which overlooks association errors, potentially leading to artificially high scores by permitting many identity switches. For biologists, HOTA is a more appropriate metric as it better reflects the importance of accurate identity tracking over time. This discrepancy is less evident in existing datasets with shorter tracks, where the likelihood of identity switches is lower.” Further, we will add a comment that HOTA is currently not suited for evaluating results for multi-classes MOT problems which could be a new direction to work towards.
>
> Q2.	Unusual re-ID
> A2.	We would like to highlight that while an unusual approach to Re-ID, we are not the first to provide a similar type of dataset. Our approach drew inspiration from research on the re-identification of cars by Wang et al., which aims to track the same vehicle across different streets with multiple cameras. In our case, similar ground truthing was not possible.
>
> Individuals in several species, including blackbuck, appear identical to the human eye, even at close distances. Understanding certain aspects of animals’ lives requires tracking their behavior over time, but preparing Re-ID datasets with rich ground truth in the wild remains a huge challenge. We offer novel ground truth to learn the appearance of the same individual from slightly different views. This problem is the first step towards solving the Re-ID problem (over days) with such challenging animals outside the scope of a single video. At this point solving this will directly enhance the study of blackbuck, and similar animals, by enabling longer individual trajectories and possibly identifying individuals that may be temporarily occluded. Our problem directly supports further ideas on multi-drone tracking for similar-looking animals in large herds. We will add this explanation in the relevant section (Sec 3.3.3) and discuss future challenges by adding a “Future Work” section.
>
> *All references used above are the same as the one used in the paper.
> We can add the proposed changes with the allotted extra space with camera-ready versions.

---

### Official Review · Reviewer_PYY4 · 2024-07-25
**Solid new wild animal tracking dataset**

**Rating:** 8
**Confidence:** 3
**Correctness:** Yes
**Clarity:** Yes

**Review:**

The authors present a significant expansion in the size and quality of multi-animal tracking datasets in the wild. I wouldn't say the work is necessarily original, but it is executed and communicated soundly and represents a significant advance for the aerial animal tracking field.

**Strengths:**

Paper is well-written, figures are clear.

Given the quality and size of the dataset, this will be the go-to dataset for this field for some time.

**Additional Feedback:**

Great work.

**Documentation:**

Yes.

**Ethics:**

No.

**Limitations:**

See above.

**Opportunities For Improvement:**

No major issues.

I do think it is probably worth discussing potential misuse of the dataset -- hunting/poaching came to mind.

Also, can the authors please expand the discussion to convey a bit more about where they think the field is heading, and what the next landmark dataset in the field will look like?

**Relation To Prior Work:**

Yes.

**Summary And Contributions:**

The authors present a large dataset of high-resolution videos of wild antelope collected via drone. In addition to the video footage, the authors provide animal-specific tracking annotations that enable benchmarking of multi-animal tracking over large areas and re-identification of the same animal in separate videos (in their case, taken concurrently in a separate drone, although the real utility would be to re-ID the same animal over days/months. As the authors explain clearly, up to this point there have been very few datasets for multi-animal tracking and re-identification. The authors remedy this and do a great job presenting and communicating their results and advances.

---

> ### Author Rebuttal · Authors · 2024-08-16
>
> We thank the reviewer for recognizing our efforts in generating this dataset and for their encouraging and positive evaluation of our work. Below, we address the few concerns raised.
>
> Q1. Potential misuse of the dataset
> A1. This is a valid concern, and we will include the following text in section 3.3.1 to address it:
>
> “The dataset provided is specifically designed for MOT within the context of blackbuck leks. It does not include images of blackbuck outside the lek, even within the same habitat. Outside this six-week breeding season, blackbuck typically moves in herds, a scenario not captured in our data. Additionally, our dataset does not account for seasonal variations in coat color, such as the lighter coloration males exhibit outside the breeding season. Given these factors, we do not anticipate misuse of our dataset in hunting or poaching contexts.”
>
> As a measure of safety, we will remove GPS information from all videos and images before uploading them publicly. It is to be noted that leks are easily visible from space in Google Maps. However, the positive aspect is that all known lekking areas are highly protected. Therefore, this dataset is rather valuable to the forest officials (our collaborators) as it will enable aerial censuses of animals within the lek—an approach previously infeasible due to a lack of appropriate methods or technology.
>
> Q2. Future directions
> A2. We agree that outlining future directions is crucial and will add a "Future Work" section in the paper, including the following points:
> Future datasets: As noted in A1, future datasets for animal MOT should account for enhancing morphological, behavioral, and environmental variation in datasets. As pointed out by the reviewer, this will introduce concerns of misuse of datasets and one will have to consider these aspects more carefully. Additionally, while our focus is the study of blackbuck, one could imagine datasets capturing interactions among a wide range of economically important species and species that are targets of conservation action. Since many of these animals are tetrapods and have similar body plans, tools developed on datasets like ours should translate to other use cases reasonably straightforwardly.
>
> Algorithms: A huge requirement of the animal behavior community is the Re-ID of animals with difficult animals. As identified by the reviewer, our current dataset focuses on re-ID when recorded concurrently by two separate drones. However, the long-term goal is to identify these individuals across days, months, and years. We presented one idea for ground truth, but we need better ideas to design ground truth for this problem.
> For MOT, we anticipate algorithms should be able to track small animals in high-density scenarios within reasonable computation capability. MOT methods need improvement in speed and reduced switches of identity, especially with position-based trackers. Our best-performing model (YOLOv8m model + BoT-SORT) has long training and inference times. This is still not practical for processing long datasets multiple times.
>
> Evaluation metrics: MOT problems for animals may require novel benchmarking metrics. We already show that MOTA is not suitable for biologists because identify switches are not penalized which results in artificial high scores. HOTA appears to be a better metric for tracking single-class objects and penalizes identity switches but it does account for multiple classes. We need novel metrics for dealing with MOT results with multiple classes, similarly, trackers can also take advantage of classes to avoid switching.
>
> If accepted, we will incorporate these details in the additional content allowed for the camera-ready version.

---

### Author Response · Authors · 2024-08-19
**Clarification on references for reviewers used in rebuttal**

Dear reviewers,

We apologize for the slight error in our rebuttal format. We have referred to reviewers based on the order of reviews "Reviewer 1, 2 etc." instead of the assigned codes. Below we provide clarification on the referenced names.

Reviewer 1 : PYY4

Reviewer 2 : odky

Reviewer 3 : TLDH

Reviewer 4 : GyVH

Thank you all for constructive feedback on our paper. We are looking forward to discussing the paper further.

---

### Author Response · Authors · 2024-08-30
**Towards the end of discussion rounds - Summary**

Dear reviewers,

Once again, the authors would like to thank you for your comments. We are happy that all reviewers unanimously agreed that the dataset is timely, novel, and soundly constructed with clear information about the problem and the process of data collection. The reviewers also encouraged our efforts in preparing such complex datasets with real animals and found the experiments sufficient and the datasets are fully documented.
Unfortunately, the first version of the paper lacked clear statements regarding the integration between computer science and biology, the Re-ID problem was defined unusually and reviewers also reported that information on the future use of the dataset for biologists and computer scientists was missing.
We have addressed these and other minor corrections in our rebuttal with specific discussion on each topic. The authors have offered changes in Section 2,3 to highlight a strong interface between biology and computer science. The authors will add a new section on future work to discuss the potential use of the dataset in computer science and biology. These changes have made the paper much clearer and stronger.

At the end of the discussion round, we hope our answers have satisfied the concerns raised by the reviewers.

Thank you.

---

### Author Response · Authors · 2024-09-04
**Further discussions**

Dear reviewers,

We thank you for your comments and believe that incorporating them in our manuscript greatly improves its quality. We hope our rebuttal letters clarify your queries and cover ways in which we incorporate your suggestions ("Opportunities for improvement" section) into our manuscript.

If there are any further questions, please do not hesitate to contact us. We're happy to have further discussions and would like to work with you to make this dataset as valuable to the community as possible.

Thank You.

---

### Decision · Program_Chairs · 2024-09-26

**Decision:**

Accept (Poster)

**Comment:**

This paper received mixed review results ranging from "5. Marginally below acceptance threshold" to "8: Top 50% of accepted papers, clear accept". As a result, the average score over the four reviewers is slightly lower than the borderline.
The dataset in this paper is clearly different from other standard benchmark datasets in artificial intelligence and machine learning. The dataset in this paper includes animal behavior records collected by flying UAVs, which is used for multi-object tracking (MOT) and re-identification (Re-ID) in wild animals. Due to the novelty of the dataset in this paper, I think that this paper is acceptable. The dataset in this paper will become useful for the problem of multi-object tracking (MOT) and re-identification (Re-ID) in wild animals, which may enhance the progress of this research field.